

# Littorally adaptive? Testing the link between habitat, morphology, and reproduction in the intertidal sculpin subfamily Oligocottinae (Pisces: Cottoidea)

Thaddaeus J. Buser[1], Michael D. Burns[1] and J. Andrés López[2,3]

[1] Department of Fisheries and Wildlife, Oregon State University, Corvallis, OR, United States of America
[2] College of Fisheries and Ocean Sciences, University of Alaska—Fairbanks, Fairbanks, AK, United States of America
[3] University of Alaska Museum, Fairbanks, AK, United States of America

Corresponding author
Thaddaeus J. Buser,
busert@oregonstate.edu

## ABSTRACT

While intertidal habitats are often productive, species-rich environments, they are also harsh and highly dynamic. Organisms that live in these habitats must possess morphological and physiological adaptations that enable them to do so. Intertidal fishes are generally small, often lack scales, and the diverse families represented in intertidal habitats often show convergence into a few general body shapes. However, few studies have quantified the relationship between phenotypes and intertidal living. Likewise, the diversity of reproductive traits and parental care in intertidal fishes has yet to be compared quantitatively with habitat. We examine the relationship of these characters in the sculpin subfamily Oligocottinae using a phylogenetic hypothesis, geometric morphometrics, and phylogenetic comparative methods to provide the first formal test of associations between fish phenotypes and reproductive characters with intertidal habitats. We show that the ability to live in intertidal habitats, particularly in tide pools, is likely a primitive state for Oligocottinae, with a single species that has secondarily come to occupy only subtidal habitats. Contrary to previous hypotheses, maximum size and presence of scales do not show a statistically significant correlation with depth. However, the maximum size for all species is generally small (250 mm or less) and all show a reduction in scales, as would be expected for an intertidal group. Also contrary to previous hypotheses, we show that copulation and associated characters are the ancestral condition in Oligocottinae, with copulation most likely being lost in a single lineage within the genus *Artedius*. Lastly, we show that body shape appears to be constrained among species with broader depth ranges, but lineages that occupy only a narrow range of intertidal habitats display novel body shapes, and this may be associated with habitat partitioning, particularly as it relates to the degree of wave exposure.

## INTRODUCTION

Intertidal habitats are often highly-productive, species rich environments (*Leigh et al., 1987*). Yet intertidal areas are also one of the harshest marine environments, often subject to rapidly changing physical conditions such as wave action, temperature, and current, as well as factors that affect homeostasis of resident organisms, such as pH and dissolved oxygen (*Davenport & Woolmington, 1981*; *Bridges, 1993*; *Martin, Lawson & Engebretson, 1996*). Fishes living in these areas often display common physical characteristics such as small size (*Gibson, 1982*) and a reduction of scales (e.g., intertidal members of Blenniidae, Gobiesocidae, Pholidae, see *Chotkowski, Buth & Prochazka, 1999*; *Knope & Scales, 2013*), presumably as means of coping with the unique set of challenges presented by intertidal habitats. Likewise, the body shapes of intertidal fishes appear constrained to take on one of only a few stereotypical shapes, such as elongate and eel-like (e.g., Pholidae), cylindrical and tapered (e.g., Cottoidea), or dorso-ventrally compressed (e.g., Gobiesocidae; for full descriptions see *Horn, 1999*).

In contrast to the somewhat predictable morphological characteristics of intertidal fishes, the reproductive biology of these species is diverse and does not show apparent patterns between intertidal and subtidal taxa (reviewed in *DeMartini, 1999*; *Coleman, 1999*). However, our understanding of many of these morphological and reproductive patterns in intertidal fishes is based only on qualitative assessments. Body shape, for instance, has never been quantitatively described and compared among or between any group(s) of intertidal fishes. A quantitative approach may shed additional light on the patterns and processes of adaptation to intertidal habitats in fishes. A phylogenetic comparative approach is one way to better understand the relationship of habitat, morphological, and reproductive characters in intertidal fishes, and the marine sculpin (family Psychrolutidae *sensu Smith & Busby, 2014*) subfamily Oligocottinae is a relatively well-studied group and excellent candidate in which to do so.

The 16 species that make up Oligocottinae are found in a variety of shallow nearshore habitats across the Pacific coast of North America (*Hubbs, 1926*; *Taranetz, 1941*; *Ramon & Knope, 2008*; *Buser & López, 2015*). The members of this subfamily occupy a range of subtidal and intertidal habitats, with varying degrees of intertidal occupation across species (*Bolin, 1944*; *Lamb & Edgell, 1986*; *Mecklenburg, Mecklenburg & Thorsteinson, 2002*). Likewise, oligocottines display a broad range of reproductive strategies ranging from copulation and internal insemination to spawning and external mixing of gametes (*Petersen et al., 2005*; *Abe & Munehara, 2009*).

Recent studies have suggested that the diversification of Oligocottinae is associated with a shift in habitat by the group (*Ramon & Knope, 2008*; *Knope & Scales, 2013*). Subtidal habitats are believed to be the ancestral condition of the subfamily and the putative shift from subtidal to intertidal habitats is thought to have been followed by relatively rapid diversification of the subfamily. The shift in habitat is associated with adaptive morphological changes, which include smaller body size and fewer scales in intertidal species when compared to their deeper-dwelling relatives (*Knope & Scales, 2013*). Habitat
specialization is thought to have occurred within the subfamily, such that the group contains intertidal and "transitional" taxa, with the intertidal taxa being the most species rich (*Ramon & Knope, 2008*).

At the heart of these results, however, is an unanswered question, namely: how does one categorize the habitat type (e.g., "intertidal") of each species? Intertidal habitats comprise a range of depths which change on daily, seasonal, and yearly cycles. Categorizing these habitats and ascribing them to a fish, which is free to move across and occupy all habitat types with every flooding tide, presents many potential pitfalls (this conundrum is reviewed in *Gibson & Yoshiyama, 1999*). The ways in which fishes use these habitats ranges from intertidal residents to intertidal transients (*Breder, 1948*; *Gibson, 1969*; *Thomson & Lehner, 1976*; *Potts, 1980*) and this continuum only further complicates the qualitative categorization of these fishes. Given these uncertainties, and the potential for arbitrary categorizations to impact the results of comparative analyses, it could be useful to take a different approach.

Many species venture into intertidal habitats during high tide but do not remain during low tides ("intertidal transients"). Conversely, some species remain in intertidal habitats throughout the tidal cycle. These "intertidal residents" are often found in special habitats during low tides, such as in tide pools or under exposed rocks, and use a suite of behavioral and physiological adaptations to cope with the challenging conditions that they present (*Martin, 1996*; *Gibson & Yoshiyama, 1999*; *Mandic, Todgham & Richards, 2009*; *Martin & Bridges, 1999*; *Evans, Claiborne & Kormanik, 1999*). The number of prerequisite adaptations needed to survive in tide pool habitats suggests that species that regularly utilize them possess at least the capacity to function as intertidal residents.

Small size (i.e., length) and a reduction of scales have been reported for many resident intertidal species and these characters show an adaptive shift between subtidal and intertidal oligocottine sculpins (*Knope & Scales, 2013*). While reproductive characters are not known to correspond to intertidal vs subtidal habitats (*Coleman, 1999*; *DeMartini, 1999*), the relationship between depth and reproductive characters has yet to be formally tested. Reproductive traits are very diverse in sculpins, particularly regarding copulation and parental care (*Abe & Munehara, 2009*). While copulation is difficult to observe directly, characters that are putatively associated with this trait, such as the presence of an enlarged genital papilla, and spermatozoon morphology, are more readily observable. Parental care is also difficult to observe in many species, but has important evolutionary implications.

In this study, we forego categorization of habitat and instead infer depth ranges for each species to test whether the host of morphological and reproductive traits putatively linked to species in intertidal habitats in fact correlate with depth. To do so, we construct a phylogenetic hypothesis of the subfamily Oligocottinae using previously published molecular sequence data and use ancestral state reconstruction and phylogenetic comparative methods to test the relationship between depth range and morphological, reproductive, and body shape characters in the group.

## MATERIALS & METHODS

### Phylogenetic framework

We constructed a phylogenetic framework using previously reported DNA sequences from all oligocottine species (sample size per species: 1–9 individuals, median 5) and several outgroups from the cottoid families (*sensu Smith & Busby, 2014*): Agonidae ($n = 6$ spp.), Cottidae ($n = 1$ sp.), Hexagrammmidae ($n = 1$ spp.), Psychrolutidae ($n = 11$ spp.), and Rhamphocottidae ($n = 1$ sp.). These outgroup taxa are consistent with the most recent phylogenetic hypotheses of broader cottoid relationships (*Knope, 2013*; *Smith & Busby, 2014*). This dataset is accessible on Genbank (accession numbers KP826911–KP827632, see Table S1) and contains sequence data from eight molecular loci: one mitochondrial protein-coding locus (Cytochrome c oxidase, COI), two nuclear introns (exon-primed intron crossing (EPIC) locus 1777E10 and EPIC locus 4174E20) and five protein-coding nuclear loci (early growth response protein 1 (EGR1); mixed-lineage leukemia (MLL); patched domain-containing protein 1 (ptchd1); Rhodopsin; and Sushi, von Willebrand factor type A, and pentraxin domain-containing 1 (SVEP)). Multiple sequence alignments (MSAs) for each locus were generated in ClustalW (*Larkin et al., 2007*). Alignments were visually inspected, trimmed, and concatenated in Mesquite v3.2 (*Maddison & Maddison, 2016*). The best fitting model of molecular evolution for each locus was identified using the Akaike information criterion (*Akaike, 1973*; *Posada & Buckley, 2004*), with the model comparison routines implemented in MrModeltest v2 (*Nylander, 2004*).

The molecular dataset contains multiple representatives for each species, so we estimated a species tree using the multispecies coalescent model (*Heled & Drummond, 2010*) in BEAST v1.8.2 (*Drummond et al., 2012*). A species set was defined, based on the results of recent phylogenetic hypotheses (*Knope, 2013*; *Smith & Busby, 2014*), for the superfamily Cottoidea, which contains all taxa included in the dataset except *Hexagrammos lagocephalus*, which was treated as an outgroup. The species of each sampled individual in the dataset was assigned as a discrete trait. These species assignments were tested and validated in *Buser & López (2015)*. For each locus, the model of molecular evolution yielding the lowest AIC value (as calculated in MrModeltest) was applied. The rate of molecular evolution was modeled as an uncorrelated lognormal relaxed clock (*Drummond et al., 2006*) and was unlinked across all loci. All tree models share a birth-death speciation tree prior with a piecewise linear and constant root population size model and a UPGMA starting tree. Four independent analyses were run for 200 million generations each and were sampled every 20,000 generations. MCMC logs were visualized using Tracer v1.6 (*Rambaut et al., 2014*) to determine convergence and an appropriate number of generations to discard as burn-in. Burn-in was removed and trees combined using LogCombiner v1.7.5 (*Drummond & Rambaut, 2007*). The phylogeny was pruned in the R statistical environment (*R Core Team, 2015*) using the "extract.clade" function from the package "ape" (*Paradis, Claude & Strimmer, 2004*) to only include members of the subfamily Oligocottinae plus the outgroup taxon *Chitonotus pugetensis*. An R script, "LitorallyAdaptiveScript.R", detailing these commands and all other operations performed in R, along with all pertinent data (phylogeny, data matrix, etc.) is available in Supplemental Information 1.

## Character coding

To understand the relationship between a species' depth preferences and its size, reproductive habits, or scale patterns, we coded the following characters from previous studies and, where possible, verified our findings by examining museum specimens and/or collection data (summarized in Table 1):

1. **Depth range (Continuous).** Collection data for all specimens of each species of Oligocottinae and the outgroup taxon *C. pugetensis* were collated from museum records from the following natural history collections: University of Alaska Museum (UAM), University of British Columbia Beaty Biodiversity Museum (UBCBBM), University of Washington Burke Museum Fish Collection (UW), Oregon State University Fish Collection (OS), California Academy of Sciences (CAS), Natural History Museum of Los Angeles County (LACM), University of Michigan Museum of Zoology (UMMZ) and Scripps Institute of Oceanography Marine Vertebrates Collections (SIO). These records were accessed through institution-specific (UW, UBCBBM, CAS) or the multi-institutional database interfaces (all others) VertNet.org, Arctos.Database.Museum, and FishNet2.org (see Table S2 for all museum records analyzed). For each species, we extracted collection depth data from all museum holdings of adult specimens for which it had been recorded. Some collection depths are recorded as a range, in these cases, we used the maximum depth in the range. Where the collection depth and/or locality is described as "tide pool," "intertidal," etc., we assigned a collection depth of 0 m. To lessen the effects of outliers, we selected a depth range (i.e., minimum depth and maximum depth) for each species that includes 95% of museum collection depths (illustrated in Fig. 1). For the purposes of this study, we will refer to this depth range as the range where each species is "commonly" collected. To verify these depth ranges, maximum and minimum depth records for each species were cataloged and cross-examined from multiple sources *Bolin, 1944*; *Miller & Lea, 1972*; *Eschmeyer, Herald & Hammann, 1983*; *Mecklenburg, Mecklenburg & Thorsteinson, 2002*; see Table S3). Where these previously published depth maxima and minima disagree, we chose the median value for each. Many of these ranges include only imprecise descriptions such as "tide pools" and "intertidal areas". In these cases, we assigned a minimum depth value of 0 m and a maximum depth value of 2 m.

2. **Tide pool occupancy (Presence, absence).** We noted which taxa were explicitly collected from tide pools in museum collection data, in previously published depth ranges, and in primary literature.

3. **Length (Continuous).** Maximum recorded length of each species was cataloged and cross-examined from multiple sources (*Bolin, 1944*; *Miller & Lea, 1972*; *Eschmeyer, Herald & Hammann, 1983*; *Mecklenburg, Mecklenburg & Thorsteinson, 2002*; *Knope & Scales, 2013*). Where sources disagreed, we used the median value.

4. **Squamation (Presence, absence).** For the purposes of this study, squamation is defined as any dermal ossification outside of the lateralis system. This includes scales, prickles, and scutes. The evolution of scale types in sculpins is poorly understood, but what is known suggests that the modified scales found in Oligocottinae may each represent an equal number of evolutionary steps away from the ancestral ctenoid scale type, with the latter not represented in any extant cottoid (*Jackson, 2003*). We therefore feel that in the

**Table 1** Matrix of characters examined and character states for each species.

| Species | Minimum depth (m)* | Maximum depth (m)* | Tide pool occupancy | Maximum length (mm) | Squamation | Enlarged genital papilla | Spermatozoon morphology | Copulation | Parental care |
|---|---|---|---|---|---|---|---|---|---|
| *Artedius corallinus* | 0 | 27 | Present | 140[3,12,13] | Present[3] | Absent[3] | ? | ? | ? |
| *Artedius fenestralis* | 0 | 52 | Present | 140[3,10,12,13] | Present[3] | Absent[3,10] | Oval[9] | Absent[9] | Present[9] |
| *Artedius harringtoni* | 0 | 18 | Present | 102[3,10,12,13] | Present[3] | Absent[3,10] | Intermediate[4,9] | Present[9] | Present[9] |
| *Artedius lateralis* | 0 | 6 | Present | 140[3,10,12,13] | Present[3,11] | Absent[3,10] | Oval[4,9,15] | Absent[9] | Present[9] |
| *Artedius notospilotus* | 0 | 20 | Present | 250[3,12,13] | Present[3] | Absent[3] | Oval[4] | ? | ? |
| *Chitonotus pugetensis* | 7 | 137 | Absent | 230[3,10,12,13] | Present[3,11] | Present[3,10] | ? | Present[1,6] | ? |
| *Clinocottus (Oxycottus) acuticeps* | 0 | 1 | Present | 64[3,10,12,13] | Absent[3] | Present[3,10] | Slender[4] | ? | Absent[14] |
| *Clinocottus (Clinocottus) analis* | 0 | 6 | Present | 180[3,12,13] | Present[3] | Present[3] | Slender[4] | Present[1,5] | Absent[1,5] |
| *Clinocottus (Blennicottus) embryum* | 0 | 0 | Present | 70[3,10,12,13] | Absent[3] | Absent[3] | Slender[4] | ? | ? |
| *Clinocottus (Blennicottus) globiceps* | 0 | 1 | Present | 190[3,10,12,13] | Absent[3] | Present[3,10] | Slender[4] | ? | ? |
| *Clinocottus (Blennicottus) recalvus* | 0 | 2 | Present | 130[3,12,13] | Absent[3] | Present[3] | Slender[4] | Present[1,7] | Absent[1,7] |
| *Leiocottus hirundo* | 0 | 26 | Absent | 250[3,12,13] | Present[11] | Present[3] | ? | ? | ? |
| *Oligocottus maculosus* | 0 | 2 | Present | 90[3,10,12,13] | Absent[3,11] | Present[3,10] | Slender[4,15] | Present[1] | Absent[1] |
| *Oligocottus rimensis* | 0 | 1 | Present | 65[3,10,12,13] | Present[3] | Present[3,10] | Slender[4] | ? | ? |
| *Oligocottus rubellio* | 0 | 8 | Present | 100[3,12,13] | Absent[3] | Present[3] | Slender[4] | ? | ? |
| *Oligocottus snyderi* | 0 | 1 | Present | 90[3,10,12,13] | Absent[3] | Present[3,10] | Slender[4,15] | Present[1,8] | ? |
| *Orthonopias triacis* | 0 | 27 | Present | 100[3,12,13] | Present[3,11] | Absent[3] | Slender[4,15] | Present[1,2] | ? |

**Notes.**

Numbered references for each state are indicated in superscript and are as follows: 1, *Abe & Munehara (2009)*; 2, *Bolin (1941)*; 3, *Bolin (1944)*; 4, *Hann (1930)*; 5, *Hubbs (1966)*; 6, *Misitano (1980)*; 7, *Morris (1952)*; 8, *Morris (1956)*; 9, *Petersen et al. (2005)*; 10, *Mecklenburg, Mecklenburg & Thorsteinson (2002)*; 11, *Jackson (2003)*; 12, *Miller & Lea (1972)*; 13, *Eschmeyer, Herald & Hammann (1983)*; 14, *Marliave (1981)*; 15, *Koya et al. (2011)*.

* Minimum and maximum depth are taken from the depth range that contains 95% of museum collection depths for each species. See Methods section and Table S4.

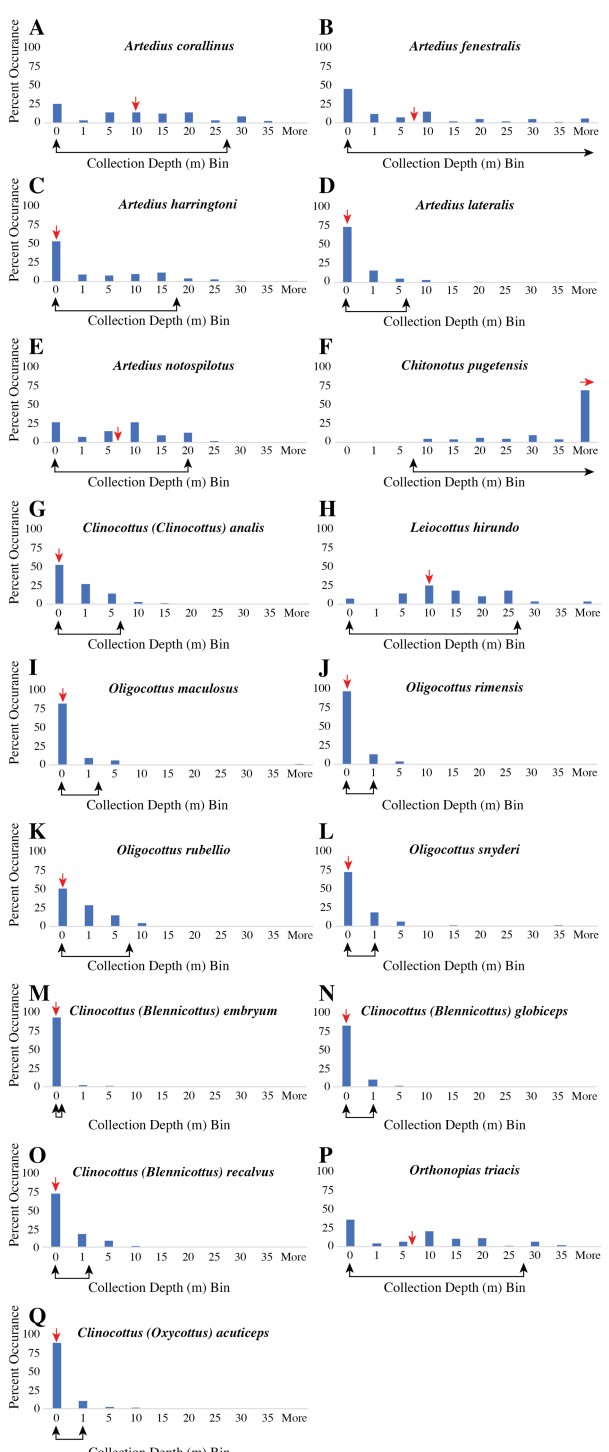

**Figure 1  Histogram of all recorded collection depth from museum records of each species of the sculpin subfamily Oligocottinae and the outgroup taxon *Chitonotus pugetensis*.** (A–E) species of the genus *Artedius*; (F) *C. pugetensis*, (G) *Clinocottus (Clinocottus) analis*; (H) *Leiocottus hirundo*; (I–L) species of the genus *Oligocottus*; (M–O) species of the subgenus *Clinocottus (Blennicottus)*; (P) *Orthonopias triacis*; (Q) *Clinocottus (Oxycottus) acuticeps*. In each panel, the *x*-axis represents bins of collection depth in meters. The first bin "0," contains only collection depths that were recorded as 0 m or where the habitat or collection depth is described as "tide pool," "intertidal," or the like.

context of this study it is unjustifiable to discriminate between scale types in oligocottines until further study indicates otherwise. Presence of squamation was coded from descriptions in the literature (*Bolin, 1944*; *Begle, 1989*; *Mecklenburg, Mecklenburg & Thorsteinson, 2002*; *Jackson, 2003*).

5. **Enlarged genital papilla (Presence, absence).** This character was coded directly from descriptions in the literature (*Bolin, 1944*; *Mecklenburg, Mecklenburg & Thorsteinson, 2002*).

6. **Spermatozoon morphology (Oval, intermediate, slender).** Character states were adapted from descriptions in the literature (*Hann, 1930*; *Petersen et al., 2005*; *Koya et al., 2011*). Slender sperm morphology is associated with internal insemination in many groups of fishes (*Mattei, 1991*). *Petersen et al. (2005)* confirmed this observation in Oligocottinae by demonstrating that spermatozoa with a slender-type morphology are active only in seawater that has been diluted to approximate the osmolality of ovarian fluid in these sculpins, while spermatozoa with oval-type morphology are active in both dilute and full-strength seawater. This suggests that slender-type spermatozoon morphology is indicative of obligate insemination, but oval-type morphology indicates the capacity for external mixing of gametes (i.e., spawning).

7. **Copulation (Presence, absence).** For the purposes of this study, copulation is defined as the transfer of sperm from a male into the ovary of a female. The presence of copulation, where known, was determined from descriptions found in the literature (*Bolin, 1941*; *Morris, 1952*; *Morris, 1956*; *Hubbs, 1966*; *Misitano, 1980*; *Petersen et al., 2005*; *Abe & Munehara, 2009*).

8. **Parental care (Presence, absence).** For the purposes of this study, egg guarding by one or both parents is considered parental care. The presence of parental care, where known, was determined from a review of behavioral descriptions from previous literature (*Morris, 1952*; *Hubbs, 1966*; *Petersen et al., 2005*; *Abe & Munehara, 2009*).

## Character mapping and ancestral state reconstruction

To visualize the inferred evolutionary history of characters, we performed ancestral state reconstruction (ASR) of each character examined in this study. For discrete characters, we used maximum likelihood (ML) with a Markov k-state 1 parameter (Mk1) model of evolution (*Lewis, 2001*), implemented in Mesquite. The evolutionary history of continuous characters was inferred using ML in the R statistical environment with functions from the package "phytools" (*Revell, 2012*; see "Character mapping and ancestral state reconstruction" section in "LitorallyAdaptiveScript.R" in Supplemental Information 1).
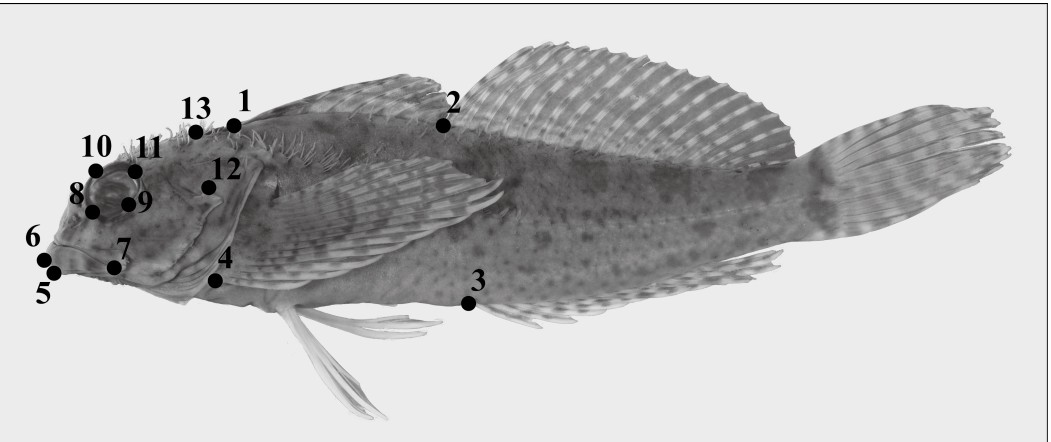

**Figure 2** Lateral photograph of *Clinocottus analis.* (OSIC 6710, 75.9 mm SL) showing thirteen homologous landmarks used to capture overall body shape in oligocottine sculpins. Landmark descriptions: (1) insertion of spinous dorsal fin, (2) insertion of soft dorsal fin, (3) insertion of anal fin, (4) ventral-most ray of the pectoral fin, (5) anterior-most tip of dentary (6) anterior-most tip of premaxilla (7) posteroventral-most tip of maxilla, (8) anteroventral-most point of orbit, (9) posteroventral-most point of orbit (10) anterodorsal-most point of orbit, (11) posterodorsal-most point of orbit (12) dorsal tip of dorsal-most preopercular spine, (13) insertion of epaxial musculature onto neurocranium.

## Body shape analysis

Qualitative assessment of body shape in intertidal fishes has not only shown differences in the shape of some intertidal species compared to their subtidal relatives (e.g., *Hypsoblennius* spp.; *Thomson & Lehner, 1976*), but also that many groups of intertidal fishes (including intertidal sculpins) have converged on a small number of stereotypical body shapes (reviewed in *Horn, 1999*). While thought provoking, these observations have yet to be tested using quantitative methods. We used landmark-based geometric morphometrics to describe and compare the body shape of each species in this study and test for correlation between body shape and the depth at which each species occurs. Body shape data were collected from digital photographs of the lateral view of museum specimens of each species. Photography followed the phototank method of *Sabaj Pérez (2009)*. To minimize the likelihood of introducing variation due to photographic artifacts (i.e., image distortion), lighting, distance to the subject, focal length, camera angle, and camera settings (e.g., aperture) were kept constant. We photographed 115 specimens, representing all 16 species in Oligocottinae plus the outgroup taxon *C. pugetensis*. Sample size per species ranges from 2 to 15 individuals, median six (Table 2). To capture overall head and body shape, landmarks were adapted from those described in previous studies of sculpin body shape (*Strauss & Bookstein, 1982*; *Strauss & Fuiman, 1985*). Preliminary analysis revealed a high frequency of distended stomachs and upturned caudal peduncles, presumably from prior feeding and preservation (respectively), so landmarks that appeared to be influenced by these variables were not included. Thirteen landmarks were ultimately used in this study (Fig. 2). We used tps-Dig2.2 (*Rohlf, 2007*) to locate the landmarks on each specimen from the digital photographs. To compare body shape across the group, landmark configurations were Procrustes superimposed using MorphoJ v1.06 (*Klingenberg, 2011*). The aligned landmark

**Table 2** Sample size (*n*) and museum lot number (Museum ID) of specimens examined for each species that was photographed for landmarking and body shape analysis.

| Taxon | *n* | Museum ID |
| --- | --- | --- |
| *Artedius corallinus* | 8 | OSIC 08140, SIO 457-34-55, SIO 057-34-55, SIO H51-34-55C |
| *Artedius fenestralis* | 9 | OSIC 05879, OSIC 09206, UW 000587, UW 017420, UW 118839 |
| *Artedius harringtoni* | 15 | OSIC 04533, OSIC 07471, OSIC 11055, UW 001011, UW 027119, OSIC 07474 |
| *Artedius lateralis* | 10 | OSIC 03175, OSIC 03178 |
| *Artedius notospilotus* | 2 | OSIC 02995, OSIC 07523 |
| *Chitonotus pugetensis* | 4 | OSIC 05269, OSIC 07016 |
| *Clinocottus acuticeps* | 7 | OSIC 06539, UAM 047689, UAM 047713 |
| *Clinocottus analis* | 5 | OSIC 06707, OSIC 06710, OSIC 08136 |
| *Clinocottus embryum* | 6 | OSIC 03009, OSIC 07071, UAM 47704 |
| *Clinocottus globiceps* | 7 | OSIC 00272, OSIC 00275, OSIC 06600 |
| *Clinocottus recalvus* | 5 | OSIC 08134 |
| *Leiocottus hirundo* | 9 | OSIC 08132, SIO 059-307-55D, SIO 071-62-55 |
| *Oligocottus maculosus* | 8 | OSIC 06628, OSIC 06663, OSIC 07467 |
| *Oligocottus rimensis* | 6 | SIO 67-151 |
| *Oligocottus rubellio* | 4 | OSIC 08133 |
| *Oligocottus snyderi* | 4 | OSIC 06541, OSIC 06668 |
| *Orthonopias triacis* | 6 | OSIC 08137 |

**Notes.**
Many museum lots contain multiple individuals.

coordinates were used to calculate a covariance matrix on which we performed a principal component analysis (PCA) in MorphoJ. The number of significant principal component axes was calculated using the broken stick method (*Frontier, 1976*; *Jackson, 1993*; *Legendre & Legendre, 2012*), implemented with the "screeplot.cca" function in the R package "vegan" (*Oksanen et al., 2017*). The significant principal component axes were used to interpret overall shape variation and visualize the distribution of species in body shape morphospace. To visually check for evidence of morphological convergence or divergence, we projected phylogenetic relatedness into the principal component morphospaces and inferred states of each significant PC axis for each ancestral node (i.e., phylomorphospace analysis; *Sidlauskas, 2008*) using the "phylomorphospace" function in the R package "phytools" (*Revell, 2012*; see "Body shape analysis" section in "LitorallyAdaptiveScript.R" in Supplemental Information 1).

## Depth correlation analysis

For both museum records and previously published depth ranges, preliminary results indicated that, while there is considerable variability in the maximum collection depth of each species in Oligocottinae, all species share a minimum recorded depth of zero meters. Given this invariability in minimum depth, we chose to use only maximum depth as our depth variable for regression analysis. We used phylogenetic generalized least squares regression (PGLS) implemented using the "gls" function in the R package "nlme" (*Pinheiro et al., 2015*) to test for a linear correlation between depth and each maximum length. To

account for potential variability in trait evolution (e.g., Brownian motion, selection, etc.), we tested three alternate single-parameter correlation structures supplied in the R package "ape" (*Paradis, Claude & Strimmer, 2004*) in each of our regression models: a Brownian motion model with correlation due to phylogenetic relatedness represented by Pagel's lambda (*Pagel, 1994*; *Pagel, 1999*), which we estimated using ML; a Brownian motion model with the rate of evolution (accelerated or decelerated) estimated using ML; and a single optimum (i.e., stabilizing selection) Ornstein–Uhlenbeck (OU) model (*Felsenstein, 1988*; *Hansen, 1997*) with the strength of attraction towards the optimum represented by alpha and estimated using ML. The best fitting model for each regression was determined by comparing AIC values. We also tested for a statistically significant difference in the average minimum or maximum depth between each state of the binary characters: presence of scales and presence of a genital papilla using phylogenetic analysis of variance (ANOVA) implemented using the "aov.phylo" function from the R package "geiger" (*Harmon et al., 2007*).

One of our hypotheses is that shallow-dwelling species show convergent morphology differing from that of their subtidal sister taxa. To test for convergent or divergent evolution of body shape, we used a stepwise model-fitting approach, "surface," that detects shifts and convergence in phenotypic optima (*Ingram & Mahler, 2013*). In this approach, each optimum contributes a parameter to an OU process of evolution. The "surface" method finds the maximum-likelihood estimate of the number and location of phenotypic optima under the OU model and collapses similar phenotypic optima together if it improves the AIC score (*Ingram & Mahler, 2013*). Phenotypic convergence is indicated when independent lineages share a common optimum. These analyses were conducted in the R environment using functions from the package "surface" (*Ingram & Mahler, 2013*).

To visualize only the aspects of body shape that covary with depth, a partial least squares analysis (PLS) was conducted on a matrix of depth variables (minimum depth and maximum depth) and Procrustes-aligned shape variables (*Rohlf & Corti, 2000*). This analysis was conducted in MorphoJ and in R using functions from the package "geomorph v2.0" (*Adams & Otárola-Castillo, 2013*). In both cases, the significance of the covariance was tested using a permutation test with 10,000 iterations (see "Correlation with depth" section in "LitorallyAdaptiveScript.R" in Supplemental Information 1).

## RESULTS

### Character mapping and ancestral state reconstruction

The trimmed, concatenated MSA dataset spans 4,695 aligned nucleotide sites, containing 1,037 variable sites. The topology of the Bayesian maximum clade credibility (MCC) phylogeny produced herein is identical to the topology of the MCC phylogeny reported in *Buser & López (2015)*, with similar levels of support for each clade (Fig. 3). As noted in *Buser & López (2015)*, this topology is similar to that of other molecular-based phylogenetic inferences of Oligocottinae (i.e., *Ramon & Knope, 2008*; *Knope, 2013*), but has substantially higher support values (i.e., Bayesian posterior probability) for many of the inferred relationships. We will use the classification and taxonomy suggested by *Buser & López (2015)* for discussion of the interrelationships of oligocottine sculpins.

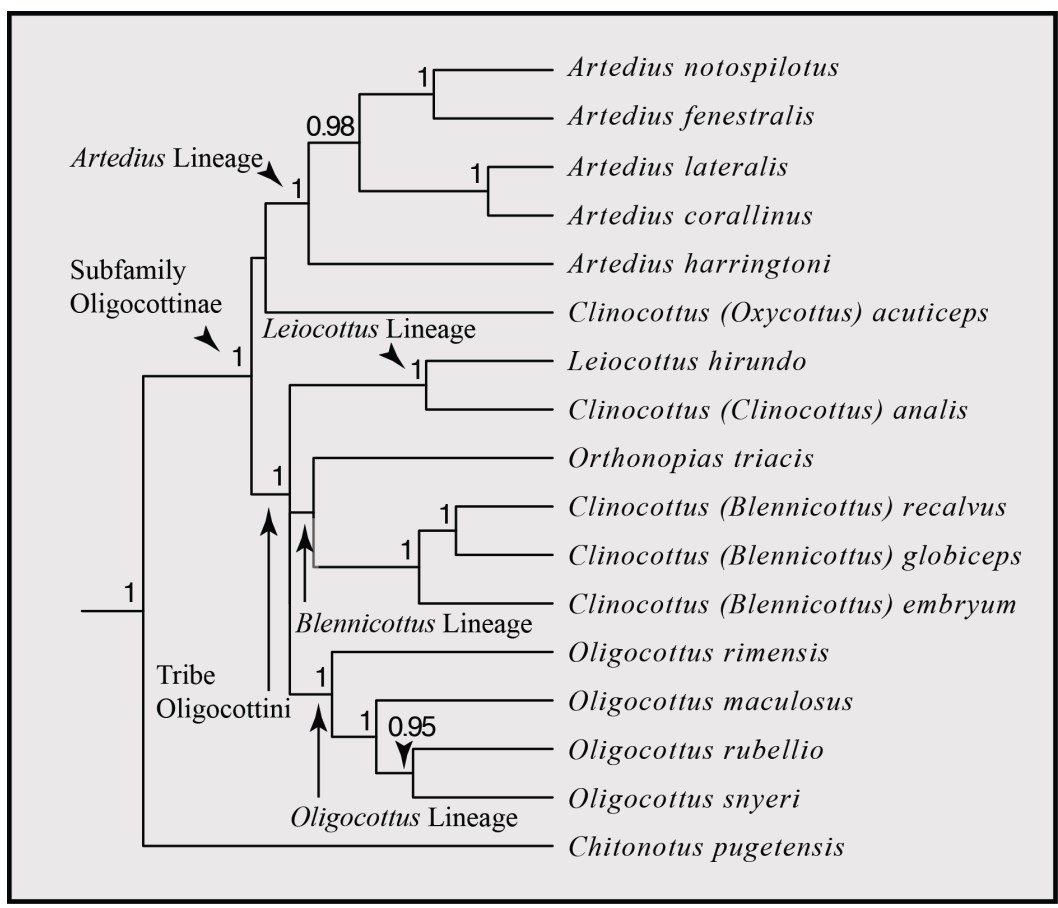

**Figure 3  Phylogenetic hypothesis of the subfamily Oligocottinae.** Phylogeny is the maximum clade credibility tree from Bayesian phylogenetic inference conducted using four independent runs of 200 million generations each using the molecular dataset published in *Buser & López (2015)*. Bayesian posterior probability scores are indicated at each node. Probabilities less than 0.50 are not displayed. The subfamily Oligocottinae, along with pertinent clades therein are labeled with arrows following the taxonomy suggested in *Buser & López (2015)*.

The outgroup taxon, *C. pugetensis*, rarely (if ever) occurs in intertidal areas (Fig. 1, Table 1, Tables S3 & S4). However, apart from *L. hirundo*, all the constituent species of Oligocottinae are regularly found in intertidal habitats and both museum records and published depth ranges include tide pools in the common collection depth or depth range data for all oligocottine species but *L. hirundo* (Fig. 1, Table 1, Tables S3 & S4). There is also explicit discussion of tide pool and intertidal occupancy for all oligocottine species except *L. hirundo* in the primary literature (Table S3). However, while the occupation of intertidal and subtidal habitats is often portrayed as an either/or scenario, there is considerable variation in the maximum depth at which each species occurs (Fig. 1, Table 1). Generally though, all oligocottine species occur at relatively shallow depths: none is commonly collected at depths greater than 55 m, most (12/16 spp.) are not commonly collected below 25 m (though there is some discrepancy between the museum collection data and the published depth ranges for *A. corallinus* and *A. fenestralis*), and four (published ranges)

to seven (museum depth data) species are common only in very shallow (i.e., 2 m depth or less) habitats (Table 1). There is considerable disagreement between the museum collection data and the published depth range for *A. notospilotus*, *C. acuticeps*, *C. analis*, and *L. hirundo*. In each case, published depth ranges indicate a maximum depth that is >10 m deeper than the depths where these species have been commonly collected in museum holdings Tables S3 & S4). However, the depth ranges are otherwise largely congruent, and the differences between the two datasets are minimal (compare depth range values in Table 1 and Table S3). All remaining analyses show identical outcomes when using either the common museum collection depth data or the previously published depth range data for each species. Given the congruence of the datasets, the indistinguishable outcome of using one over the other, and the more verifiable nature of the museum collection records, we present the results of the remaining analyses using only the common museum collection depth range of each species.

Predictably, the ASR of minimum depth shows that the most recent common ancestor (MRCA) of Oligocottinae likely occurred in shallow habitats (ML estimate: 1 m; 95% confidence interval: 0 m, 2 m). Ancestral state reconstruction of tide pool occupancy shows that with extremely high proportional likelihood (0.9988) the MRCA of Oligocottinae occurred in tide pools. In fact, even the MRCA of the *Leiocottus* lineage was likely (0.9215 proportional likelihood) capable of living in tide pools (Fig. S1). Thus, the absence of tide pool occupation in *L. hirundo* likely represents a derived state. The ASR of maximum depth suggests that the MRCA of Oligocottinae occurred down to only moderate depths (ML estimate: 23 m; 95% confidence interval: 2 m, 44 m; see Fig. 4) and suggests that the habitation of only very shallow-water habitats (maximum depth = 2 m or less) seen in *Oligocottus maculosus*, *O. rimensis*, and *O. snyderi* and in all members of the subgenus *Clinocottus* (*Blennicottus*) represents a derived state (see Table 1, Fig. 4). However, given the uncertainty of the ML estimates of maximum depth at each node (Fig. 4), and the uncertain phylogenetic relationships of *Blennicottus*, *Leiocottus*, and *Oligocottus* lineages (Fig. 1), it is not possible to claim with confidence the number of transitions that may have occurred within the subfamily.

Maximum length shows no obvious relationship with depth in Oligocottinae (illustrated in Fig. 4). All species (including the outgroup) are relatively small (none longer than 250 mm), most (12/16 spp.) do not grow longer than 150 mm, and seven species do not grow longer than 100 mm (Table 1). The ASR of maximum length suggests that the MRCA of Oligocottinae was small (132 mm, 95% confidence interval: 83 mm, 182 mm), but like maximum depth, the uncertainty of the ML estimates at each node precludes additional inference into the diversification of this trait (Fig. S2).

Squamation is common among members of Oligocottinae but is completely absent in two clades: the subgenus *Clinocottus* (*Blennicottus*), and the clade in *Oligocottus* made up of *O. maculosus*, *O. snyderi*, and *O. rubellio* (Table 1, Fig. S3). Predictably, the ASR shows that the presence of squamation is the most likely state for the MRCA of Oligocottinae (proportional likelihood: 0.81), and that the absence of scales represents an independent loss of the trait in the MRCA of each of the above clades (Fig. S3). It should be noted that we inferred the evolution of squamation using an Mk1 model, which assumes that all potential

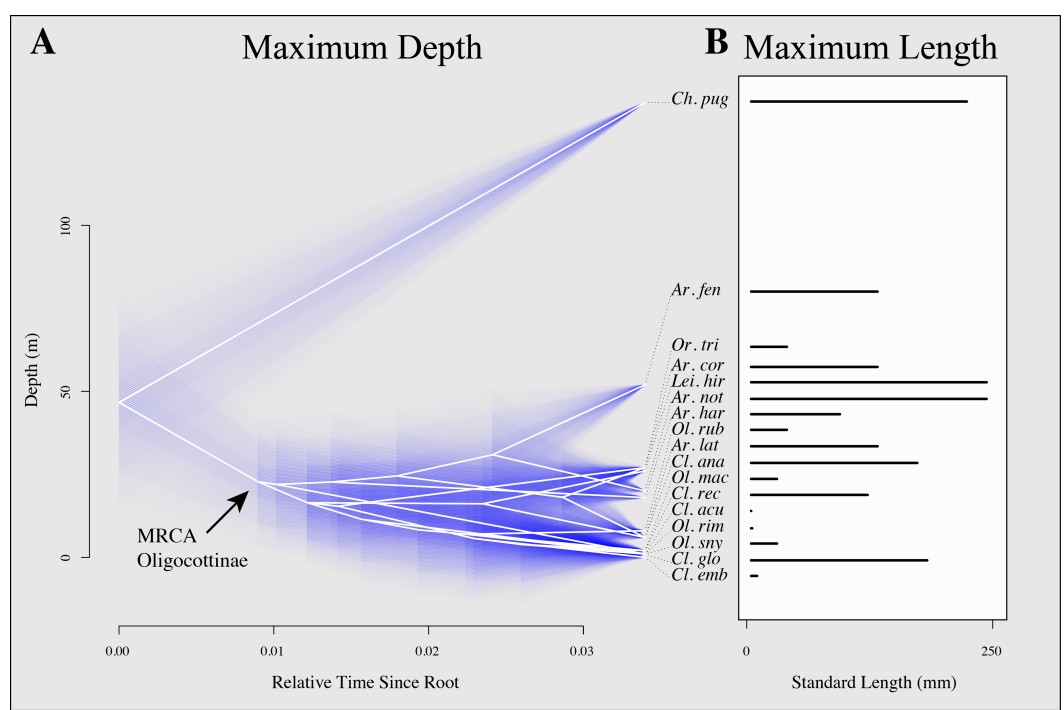

**Figure 4  Inferred evolutionary history of maximum depth and maximum length of oligocottine species.** (A) contains a phenogram showing the inferred evolutionary history of maximum depth. The maximum standard length of each species is indicated by the length of its bar in (B). Phylogenetic relationships are represented by white edges and bifurcation points represent inferred speciation events. Phylogenetic topology is from Bayesian MCC tree shown in Fig. 3. Relative time is indicated on the horizontal axis and depth in meters indicated on the vertical axis. The tips and nodes of the phylogeny are positioned on the vertical axis to reflect the maximum depth or inferred maximum depth (respectively) of each taxon. Ancestral states for each node were inferred using maximum likelihood and 95% confidence intervals for each state are represented with blue lines. Species names are abbreviated as follows: *Artedius corallinus* = Ar. cor., *A. fenestralis* = Ar. fen, *A. harringtoni* = Ar. har., *A. lateralis* = Ar. lat., *A. notospilotus* = Ar. not., *Chitonotus pugetensis* = Ch. pug., *Clinocottus (Oxycottus) acuticeps* = Cl. acu., *Clinocottus (Clinocottus) analis* = Cl. ana., *Clinocottus (Blennicottus) embryum* = Cl. emb., *C. (B.) globiceps* = Cl. glo., *C. (B.) recalvus* = Cl. rec., *Leiocottus hirundo* = Li. hir., *Oligocottus maculosus* = Ol. mac., *O. rimensis* = Ol. rim., *O. rubellio* = Ol. rub., *O. snyderi* = Ol. sny., *Orthonopias triacis* = Or. tri.

changes in state are equally probable (*Lewis, 2001*). Given the lack of rigorous study of scale evolution in sculpins, specifying a more complex model is not warranted. However, it is our opinion that re-acquisition of squamation is an extremely unlikely evolutionary scenario in oligocottine sculpins (i.e., less probable than the loss of squamation) and thus the proportional likelihood of the presence of scales for the ancestral nodes in Oligocottinae should be taken as a conservative estimate.

An enlarged genital papilla is found in all but five species of Oligocottine sculpins (Fig. 5). The ASR shows with high proportional likelihood (>0.98) that this character was present at all ancestral nodes except those within the clade composed of the members of the genus *Artedius*. Within *Artedius*, an enlarged genital papilla was likely lost in the MRCA of the clade containing *A. corallinus*, *A. fenestralis*, *A. lateralis*, and *A. notospilotus* (Fig. 5).

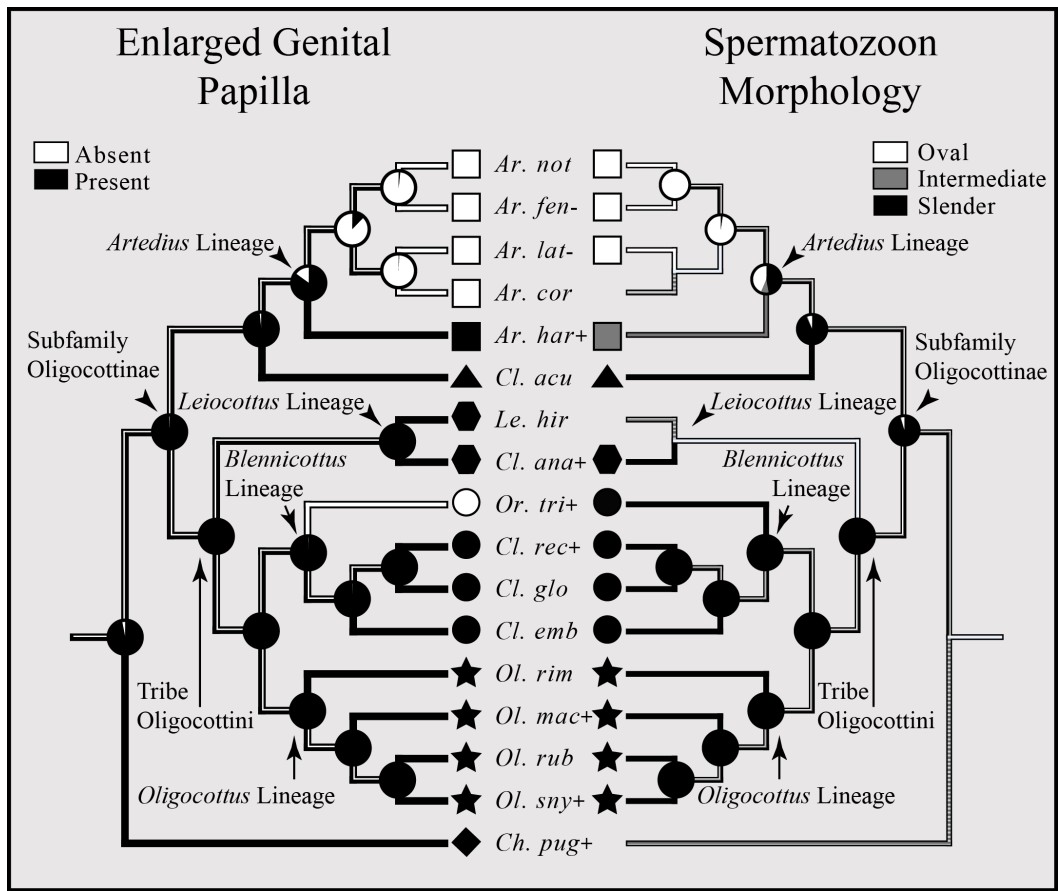

**Figure 5** **Bayesian MCC phylogeny of Oligocottinae with inferred evolutionary histories of the presence of an enlarged genital papilla and spermatozoon shape.** Where known, the presence of copulation in extant species is indicated by a "+" symbol following the abbreviated species name. Absence of copulation is indicated by a "−" symbol. Unknown states are indicated by the absence of a symbol. For the ancestral state reconstructions of the presence of an enlarged genital papilla and spermatozoon morphology, the proportional likelihood of each character for the ancestor of a given clade is depicted with a pie chart at each respective node. This scheme is also depicted on the branches between each node. Symbols at the tips of each phylogeny are indicative of the phylogenetic lineage of each species. Absence of a symbol at a tip indicates an unknown state. Squares represent the *Artedius* lineage, a triangle represents the lineage composed solely of the species *Clinocottus* (*Oxycottus*) *acuticeps*, hexagons represent the *Leiocottus* lineage, circles represent the *Blennicottus* lineage, stars represent the *Oligocottus* lineage, and a diamond represents a lineage composed solely of the species *Chitonotus pugetensis*. Species names are abbreviated as in Fig. 4.

An independent loss of the enlarged genital papilla occurred in the oligocottinin species *Orthonopias triacis*.

The distribution of spermatozoon morphology follows a pattern similar to that of the enlarged genital papilla. Outside of the clade composed of members of the genus *Artedius*, slender-type spermatozoa are present in all Oligocottine sculpins and, with high proportional likelihood (>0.90), this is the state at all ancestral nodes (Fig. 5). Within *Artedius*, an oval-type spermatozoon likely evolved in the MRCA of the clade containing *A. corallinus*, *A. fenestralis*, *A. lateralis*, and *A. notospilotus*. *Artedius harringtoni* possess an intermediate spermatozoon morphology and is the only oligocottine to do so (Fig. 5).

With one notable exception, this shows that all species with a slender-type spermatozoon morphology (which in other species is known to function only in ovarian fluid) also possess an enlarged genital papilla, which is presumably used in copulation. The exception to this observation is the species *Orthonopias triacis*, which does not possess an enlarged genital papilla, but does possess a slender-type sperm morphology.

Though not known for all species in Oligocottinae, the distribution of copulatory behavior closely follows that of spermatozoon morphology, where copulating species possess either slender or intermediate-type spermatozoon morphology, and non-copulating species possess only oval-type spermatozoon morphology (illustrated in Fig. 5; see also Fig. S4). The ASR of this character shows that, with high proportional likelihood (>0.95), presence of copulation is the likely state for all ancestral nodes outside of the clade composed of the members of the genus *Artedius*. Within *Artedius*, copulation was likely lost in the MRCA of the clade containing *A. corallinus*, *A. fenestralis*, *A. lateralis*, and *A. notospilotus*. This finding reinforces the observations of *Mattei (1991)* and *Petersen et al. (2005)* who each show that slender-type sperm morphology is indicative of copulating species. Once again, *Orthonopias triacis* presents a noteworthy case as there is evidence of copulation for the species and the species possesses a slender-type spermatozoon morphology, yet the species lacks an enlarged genital papilla or other known intromittent organ (Fig. 5).

The presence or absence of parental care has been described in less than half of all oligocottine species (6/16 spp.), but follows a similar pattern to those seen in other reproductive characters in the group, where members of the genus *Artedius* tend to differ from all other species. In this case, parental care is observed only in members of *Artedius* (Fig. 6). The ASR shows that parental care was likely present in the MRCA of *Artedius*, while a lack of parental care is the most likely state for the MRCA of the tribe Oligocottini. However, given the substantial amount of missing data for this trait, the ASR is subject to change with the addition of new observations.

## Body shape analysis

Observed body shape variation was captured by two significant principal components, which cumulatively describe 70% of the total variance. We used an outline of a specimen of *Clinocottus analis* to visualize shape change represented by each PC axis in MorphoJ (Fig. 7). Principal component (PC) 1 (52% of total observed variance) describes antero-posterior elongation/compression of the head and mouth as well as the relative size of the eye. Principal component 2 (19% of total observed variance) captures dorso-ventral elongation/compression of the body, the shape of the eye, and the slope of the snout.

There is no clear evidence of a consistent relationship between the minimum depth or maximum depth of a species and its morphology, nor evidence of morphological convergence among shallow or deep-ranging species (Fig. 8). Species with deeper ranges appear to be constrained to a common morphospace, while species that inhabit only shallow depth ranges (e.g., *Oligocottus* spp., *Clinocottus* (*Blennicottus*) spp.) appear to occupy novel and distinct areas of morphospace (Fig. 8). This observation is supported by the results of the "surface" analysis, which inferred three optima for body shape in the morphospace described by the significant PC axes: one for members of the genus *Oligocottus*

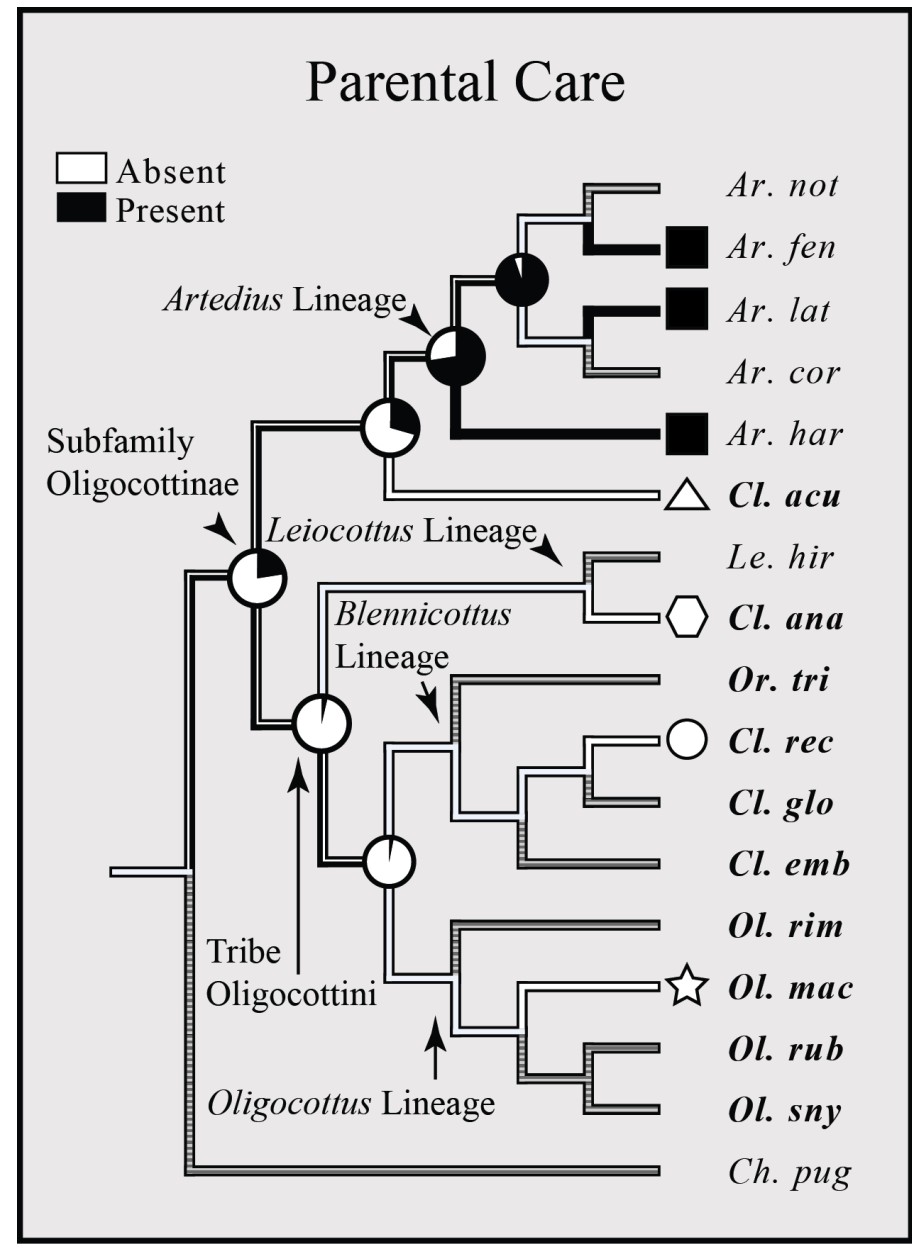

**Figure 6  Bayesian MCC phylogeny of Oligocottinae with distribution and inferred evolutionary history of parental care.** The proportional likelihood of each character for the ancestor of a given clade is depicted with a pie chart at each respective node. Symbols at the tips of the phylogeny are indicative of the phylogenetic lineage of each species, as in Fig. 4. Absence of a symbol or pie at a tip or node (respectively) indicates an unknown state. Species abbreviations in bold indicate a slender-type spermatozoon morphology present in that species.

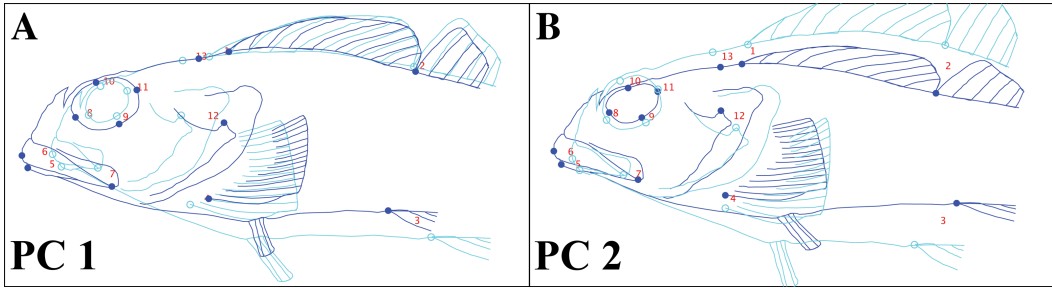

**Figure 7** **Body shape change in oligocottine sculpins represented by each of the two significant principal component axes.** (A) shows shape change captured by the first principal component (PC1); (B) shows shape change captured by the second principal component (PC2). Landmark locations are indicated by circles and are numbered as in Fig. 1. Outline sketched from the lateral photograph of *Clinocottus analis* (OSIC 6710, 75.9 mm SL) in Fig. 2. Light blue and open circles shows displacement of landmarks and interpolated warping of the outline at a value of −1 on each PC axis. Dark blue and closed circles show displacement of landmarks and interpolated warping of the outline at a value of +1 on each PC axis.

plus *Orthonopias triacis*, one for members of the subgenus *Clinocottus* (*Blennicottus*), and a third that is occupied by all other oligocottines (Fig. S5). However, there is no clear pattern in terms of the direction of the divergence in morphospace of these taxa. Interestingly, two of the optima are occupied almost entirely by taxa that are found exclusively in shallow water (i.e., *Clinocottus* (*Blennicottus*) and all but one species of *Oligocottus*), while the remaining optimum is made up almost exclusively of deeper-ranging taxa (the exception being *C.* (*O.*) *acuticeps*. Here again *Orthonopias triacis* is remarkable in that it is a deeper-ranging species, but appears to be drawn to the phenotypic optimum occupied otherwise exclusively by the genus *Oligocottus*.

## Depth correlation analysis

No morphological, reproductive, and body shape variables examined in this study show a significant correlation with maximum depth. The phylogenetic generalized least squares regression showed no significant linear correlation between depth and maximum length, and the phylogenetic ANOVA showed no statistically significant difference in average maximum depth between the binary character states of: presence of squamation ($p$-value > 0.15) or presence of an enlarged genital papilla ($p$-value > 0.70). Likewise, the results of the PLS analysis were identical in MorphoJ and in R, and failed to show a statistically significant correlation between body shape and depth range ($p$-value > 0.05). However, the analysis did reveal a few interesting observations: species that are restricted entirely to shallow water tend to display greater morphological diversity than those that range into deeper water, but generally have smaller eyes, smaller mouths, terminal placement of the mouth, and more robust bodies, while deeper-ranging species tend to have larger eyes, larger mouths, subterminal placement of the mouth, and dorso-ventrally compressed bodies (Fig. 9).

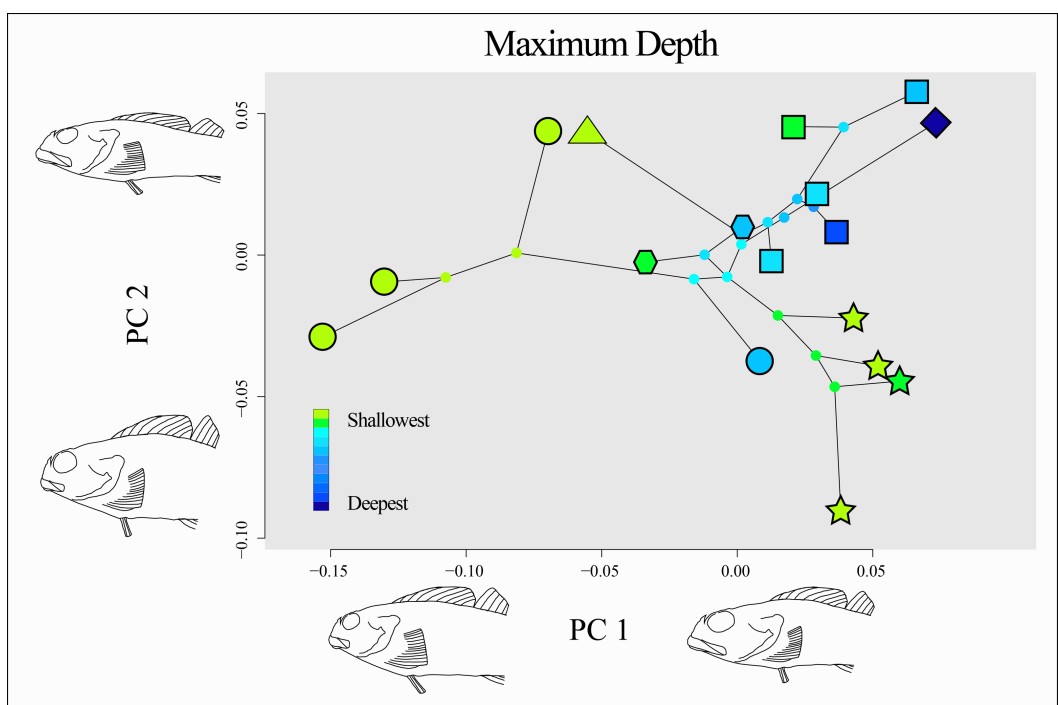

**Figure 8 Phylomorphospace of the two significant principal components of body shape in the subfamily Oligocottinae.** Outlined symbols at the tips of each phylogeny are indicative of the phylogenetic lineage of each species as in Fig. 5, ancestral nodes are indicated by small circles that are not outlined. Colors of each tip and internal node are indicative of the maximum depth of the species or the maximum likelihood (ML) estimate of the ancestral state of maximum depth, respectively. The shape change described by each PC axis is as shown as in Fig. 5. The depth data values are provided in Table 1.

# DISCUSSION

The results of our study show support for some previous hypotheses of the diversification of Oligocottinae and the general evolution of intertidal fishes in that, excepting *L. hirundo*, all oligocottines regularly occur in tide pools, and all show small bodies and few scales. However, our results do not support previous hypotheses regarding the evolution of reproductive modes in oligocottines or, to a certain extent, sculpins in general. In some cases, our conclusions directly oppose those made by previous authors. We discuss these results and some plausible explanations for our congruous and incongruous conclusions below.

## Squamation, length, and depth

The subfamily Oligocottinae should be thought of as a clade of intertidal-occurring fishes and the ability to live in intertidal depths and specialized intertidal habitats such as tide pools is likely the ancestral state of the group. However, this ability to live in tide pools does not preclude residency in other habitat types within the same species, as many of the extant and ancestral species are capable of living in a variety of depths in addition to the intertidal ones. Thus, the diversification of Oligocottinae should be characterized as

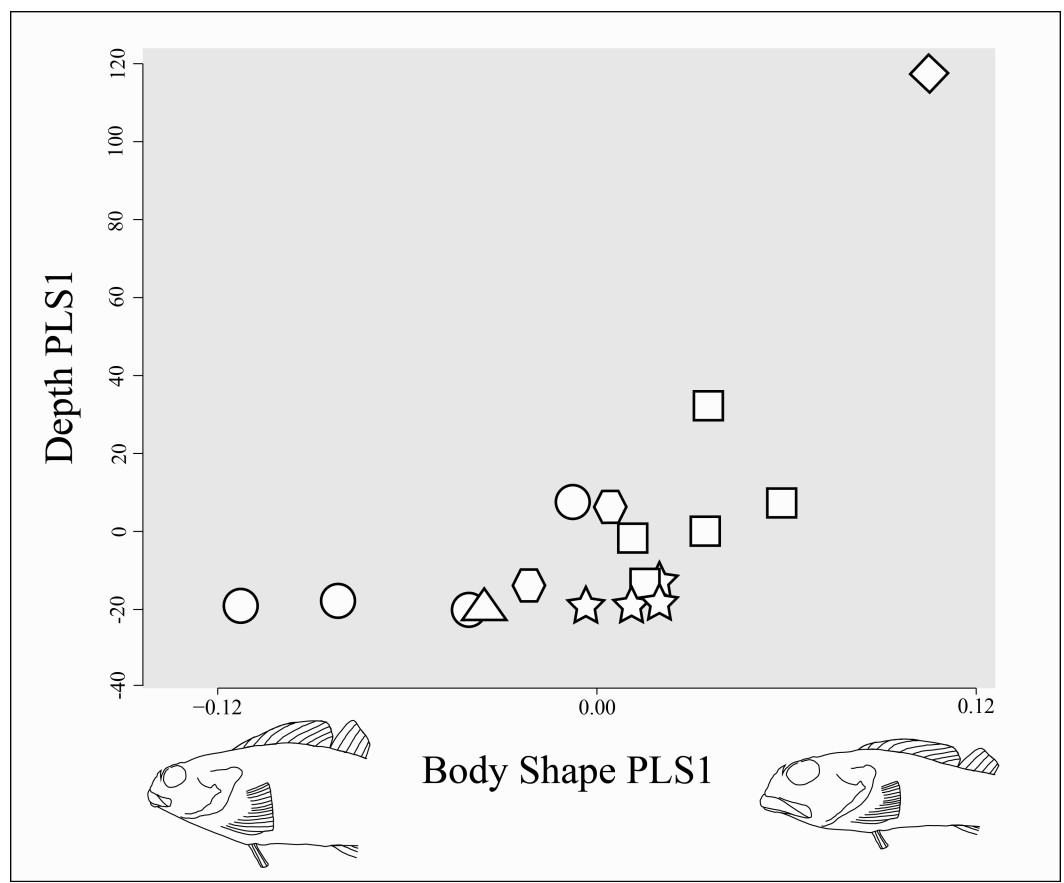

**Figure 9 Body shape change associated with change in depth range.** Body shape is represented by the average shape variables (Procrustes-aligned landmark coordinates, illustrated in Fig. 2) for each species. Depth range is represented by minimum and maximum collection depth from museum specimens of each species (see Table 1). Partial least squares (PLS) axis 1 of body shape is indicated on the horizontal axis with body shape change described by the axis shown through a warped outline sketched from the lateral photograph of *Clinocottus analis* (OSIC 6710, 75.9 mm SL) in Fig. 2. The outline on the negative side of the axis shows body shape associated with a value of −1 on PLS axis 1, the outline on the positive side of the axis shows body shape associated with a value of +1 on PLS axis 1. The PLS axis 1 of depth range is indicated on the vertical axis. Symbols are indicative of the phylogenetic lineage of each species, as in Fig. 5.

occurring within a habitat range that includes both subtidal and intertidal habitats. This may explain the general lack of correlation between depth the other characters examined in this study.

Small maximum size and a reduction in scales have been noted as common features of intertidal fishes by previous authors (*Gibson, 1982*; *Knope & Scales, 2013*), and while we found no evidence to support these hypotheses within Oligocottinae, oligocottines as a whole may in fact offer support. All oligocottines are small (none longer than 250 mm), and all show a reduction in scales when compared to a "typical" scaled member of Cottales, such as *Oxylebius pictus* (see *Jackson, 2003*). Outside of the lateral line, the most heavily scaled oligocottines possess only modified scales in a narrow band on the body along each side of the dorsal fins and on the dorsal surfaces of the head and caudal peduncle (e.g., *Orthonopias triacis*; see descriptions in *Bolin, 1944*; *Jackson, 2003*). Others possess highly

reduced scales in the form of prickles (e.g., *Clinocottus (Clinocottus) analis*; see description in *Bolin, 1944*). It is presumed that the primary reason scales are often reduced in intertidal fishes is to promote cutaneous respiration, which is dependent on well vascularized skin that is free from obstructions (*Feder & Burggren, 1985*; *Martin & Bridges, 1999*). Within Oligocottinae, it may simply be the case that the highly-reduced scales seen in the group do not cover enough surface area to interfere with cutaneous respiration in a meaningful way. Or perhaps that cutaneous respiration is restricted to only certain areas, such as the head (as seen in *Coryphoblennius galerita*, see *Zander, 1972*), or buccal chamber (reviewed in *Bridges, 1993*). This may explain the presence of scales within the group, and the fact that even the most heavily scaled members of Oligocottinae occur intertidally and in tide pools (Fig. 2). Interestingly, the one oligocottine species that does not occur intertidally (*L. hirundo*) possess only a few highly reduced scales in the form of a scattered patch of prickles located just posterior to the base of the pectoral fins (see *Jackson, 2003*). We interpret this as evidence that a reduction in scales in an ancestral condition for *Leiocottus*, and possibly Oligocottinae as a whole.

The evolutionary history of maximum size, depth range, and the presence of squamation all support the idea that the subfamily Oligocottinae is a primitively intertidal group. All species in this subfamily display conditions of these traits that are common in intertidal taxa, and all but one species are commonly found intertidally and explicitly in tide pools. These findings support those of previous studies which found Oligocottinae to represent an intertidal radiation (*Ramon & Knope, 2008*; *Knope & Scales, 2013*).

## Body shape and depth

While the body shape of deeper-ranging species is relatively conserved, groups that occupy only intertidal areas appear to be exploring novel areas of morphospace (Fig. 8, Fig. S5). This is most pronounced in members *Oligocottus* and the subgenus *Clinocottus* (*Blennicottus*), where the two groups each occupy a morphological optimum that is distinct from each other and from all but one other oligocottine (i.e., the enigmatic *O. triacis*). The morphological distinctiveness of *Clinocottus* (*Blennicottus*) is readily apparent, even to the casual observer, with antero-posteriorly compressed heads and highly robust bodies and fins (see illustrations in *Bolin, 1944*). The constituent species of this subgenus occur most abundantly in steep, rocky habitats with high wave exposure and are often the only oligocottine species found in these areas (T. Buser pers. obs.; *Green, 1971*; *Yoshiyama, 1981*; *Yoshiyama, Sassaman & Lea, 1986*; *Mgaya, 1992*, see also descriptions in *Eschmeyer, Herald & Hammann, 1983*; *Lamb & Edgell, 1986*; *Mecklenburg, Mecklenburg & Thorsteinson, 2002*). The blunt heads and short, stocky bodies of these species are also seen in other intertidal fishes occupying similarly exposed, rocky habitats and may reflect a common evolutionary response to the physical demands of living in such areas (*Kotrschal, 1988*; *Kotrschal, 1989*; *Thomson, Findley & Kerstitch, 2010*, reviewed in *Kotrschal, 1999*). If the diverse morphologies seen among intertidal specialist clades are reflective of their respective habitat partitions, it may also be the case that the constrained morphologies seen in deeper-ranging taxa reflect a kind of stabilizing selection of generalist traits that are optimal for occupying a comparatively wide variety of habitats. The relatively high

diversity of morphotypes seen in shallow vs deep-ranging species may mask morphological traits that are in fact associated with depth, as is suggested by the results of our PSL analysis (Fig. 9). However, the lack of statistical significance of this trend warrants caution on interpretation of this finding until the question can be revisited with additional taxa.

## Reproduction

While reproductive characters do not show any correlation with depth range in Oligocottinae, the evolution of these traits in the subfamily may offer new insight into the evolution of reproductive modes in cottoids. Copulation in oligocottines is associated with an enlarged genital papilla and a slender-type spermatozoon morphology. These traits are broadly distributed in Oligocottinae and are likely the ancestral state of the subfamily (Fig. 5). Importantly, the absence of copulation and associated traits in most members of the genus *Artedius* represents a loss and is thus a derived state. This finding runs counter to previous hypotheses of the evolution of reproductive modes in sculpins, which interpreted the seemingly scattered distribution of copulation in cottoids as indicative of parallel or convergent evolution of copulation from non-copulating ancestors (*Abe & Munehara, 2009*; *Muñoz, 2010*). Under this paradigm, *Petersen et al. (2005)* suggests that the ability of the oval-type sperm morphology (uniquely capable of functioning well in seawater and ovarian fluid, seen in non-copulating members of *Artedius*) to function in ovarian fluid represents a derived condition and concludes that the presence of this trait in most members of *Artedius* represents an evolutionary step *towards* copulation in the group. We conclude the opposite of *Petersen et al. (2005)*, and suggest that rather than the sperm's ability to function in ovarian fluid, it is in fact the sperm's ability to function in seawater that is a derived state and this, along with the loss of an intromittent organ, represents an evolutionary step *away* from copulation within *Artedius*.

A reduction or loss of the enlarged genital papilla is seen in other oligocottines as well. Critically, however, these species maintain a slender-type spermatozoon morphology and, where known, copulation. For example, while most oligocottines possess genital papillae that are quite large and robust, males in the genus *Oligocottus* possess papillae that are uniquely small, gracile, and thread-like (Fig. S6). Also unique to the males of this genus are modifications of the anterior portion of the anal fin (Fig. S7) which, where known, is used for grasping females during copulation (*O. snyderi*; *Morris, 1956*). It is possible that the added security and stability during copulation provided by the prehensile anal fin rays has rendered the large genital papilla seen in other oligocottines redundant. The other example of a reduction in the size of the male genital papilla is seen in *Orthonopias triacis*, where males lack an intromittent organ altogether, yet also possess slender-type sperm morphology and are known to copulate. Males of this species possess enlarged pelvic fins that face inwardly "palm to palm," and project postero-ventrally from a "pedunculated" base (*Bolin, 1944*). Perhaps these highly modified, sexually dimorphic pelvic fins are used in a grasping manner that, like in *Oligocottus*, is used to aid in copulation and has eliminated the need of a large, robust male genital papilla. Copulation without the use of an intromittent organ is seen in at least one other member of Cottales, the sea raven (*Hemitripterus villosus*). In this species, males are not known to possess any putative

grasping organs. Instead, the female everts her genital tract, which is covered in mucus, and the male ejaculates onto it, whereby the sperm become entrained in the mucus and enter the female when she inverts her genital tract (*Munehara, 1996*).

Our results show that while the presence of an enlarged male genital papilla is a likely indicator of copulation, the absence of an intromittent organ does not necessarily indicate the absence of copulation. Furthermore, our results show that non-copulating species may evolve from copulating ancestors. Given the widespread distribution of copulation and/or an enlarged genital papillae within Cottoidea (*Abe & Munehara, 2009*; *Muñoz, 2010*), we suggest that copulation and associated traits may have evolved much earlier in cottoids than has been previously estimated. Perhaps the seemingly disparate distribution of copulation in cottoids is not due to many independent evolutions of copulation, but rather to a single early evolution of copulation and multiple subsequent losses of the trait. Given the suite of complex physiological and behavioral traits associated with copulation in sculpins (e.g., internal gamete association with delayed fertilization, see *Munehara, Takano & Koya, 1989*; *Munehara, Takano & Koya, 1991*; *Munehara et al., 1997*; *Petersen et al., 2005*), the independent loss of copulation by certain lineages would, in our opinion, be a far simpler explanation for the modern distribution of the trait than would the independent evolution of copulation and all associated characters.

## Parental care

Like other reproductive traits, the distribution of parental care in Oligocottinae does not appear to be related to the distribution of depth ranges. Rather, only members of the genus *Artedius* display parental care. Strong phylogenetic signal of parental care has been reported for other groups as well (reviewed in *Coleman, 1999*), but this does not provide a satisfying explanation for why *Artedius* differs from all other oligocottines in this trait. It is interesting to note that parental care shows an almost inverse distribution to oval-type sperm morphology (i.e., obligate copulation; see Figs. 5 and 6), but the limited sample size and degree of missing data for parental care make this a tenuous connection. Many other non-copulating sculpins also display parental care (e.g., *Hemitripterus* spp., *Enophrys bison*, *Myoxocephalus* spp., *Cottus* spp.), but this trait is also seen in some copulating species, including *Artedius harringtoni* (*Abe & Munehara, 2009*). Clearly, more research is needed to better understand the evolution of parental care in cottoids, and its relationship with other aspects of their complex reproductive biology.

## CONCLUSIONS

Considering the depth ranges rather than previously published habitat categorizations of oligocottine sculpins shows substantial overlap of almost all species in intertidal habitats. This understanding of the group agrees with our findings that all oligocottine sculpins are relatively small and bear relatively few scales, two common attributes of intertidal fishes. This finding also helps to explain why the maximum depth of the common depth range does not correlate with most of the characters examined in this study, as we would expect them to vary with depth only if we are comparing intertidal fishes with subtidal fishes, and for the most part we are not. While body shape does not significantly correlate with the

maximum common collection depth, the body shape of most species with broader depth ranges appear constrained to what we interpret as a generalist morphology, while most groups that inhabit a narrow, wholly-intertidal depth range appear to have unique body shapes, perhaps suited to their specialized habitat partitions. Likewise, we find no evidence of an association between maximum depth and reproductive characters, but we do find that the evolution of these characters has likely proceeded from a primitive condition of obligate copulation using an intromittent organ to a derived state of spawning and/or the loss of an intromittent organ. This sequence is the opposite direction of that inferred by previous authors, but is clearly supported by the distribution of reproductive traits across our phylogenetic hypothesis of the group.

## ACKNOWLEDGEMENTS

Specimens were generously loaned by Katherine Maslenikov and Ted Pietsch (Burke Museum), HJ Walker (Scripps Institution of Oceanography) and Dave Catania (California Academy of Sciences). We thank Brian Sidlauskas for his advice and mentorship through much of the development and execution of this project. We thank the academic editor and two peer-reviewers of this manuscript who provided substantial feedback and helped to improve the quality and clarity of the work. We thank Matthew A. Campbell, Anne Beaudreau, Derek Sikes, Doug Markle, Jay Orr, Benjamin W. Frable, and Sophie Pierszalowski for contributing considerable technical support, philosophical discussion, and methodological expertise during various phases of this study.

### Funding

Financial support for this work was provided by NSF grant DEB 0963767 to JA López and a NSF CASE GK-12 Fellowship, Oregon State University Provost's Distinguished Graduate Fellowship, Munson Wildlife Graduate Scholarship, Oregon Council Federation of Fly Fishers Scholarship, MA Ali Graduate Chair Award in Fisheries Biology, and Carl Bond Memorial Scholarship to TJ Buser. The funders had no role in study design, data collection and analysis, decision to publish, or preparation of the manuscript.

### Grant Disclosures

The following grant information was disclosed by the authors:
NSF: DEB 0963767.

### Competing Interests

The authors declare there are no competing interests.

### Author Contributions

- Thaddaeus J. Buser and Michael D. Burns conceived and designed the experiments, performed the experiments, analyzed the data, contributed reagents/materials/analysis tools, wrote the paper, prepared figures and/or tables, reviewed drafts of the paper.

- J. Andrés López conceived and designed the experiments, contributed reagents/materials/analysis tools, wrote the paper, prepared figures and/or tables, reviewed drafts of the paper.

## Data Availability

The raw data has been supplied as a Supplementary File.

## Supplemental Information

Supplemental information for this article can be found online at http://dx.doi.org/10.7717/peerj.3634#supplemental-information.

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
