# Peer review of "Littorally adaptive? Testing the link between habitat, morphology, and reproduction in the intertidal sculpin subfamily Oligocottinae (Pisces: Cottoidea)"

_PeerJ, doi:10.7717/peerj.3634_

## Round 0.1 · original submission · Major Revisions

I now have 2 reviews back, and while both are relatively positive about the data and the intention and findings of the study, one referee feels that it suffers from a critical flaw in how species are assigned to intertidal habitat affinity. As the referee points out, "Figure 2 clearly refutes your assertion that the Oligocottinae is a primitively and overwhelmingly intertidal group." They take issue with assignment of species to a habitat affinity based only on their minimum extreme reported depth, and I find myself in agreement with the referee that a more robust treatment of how species are assigned to habitat would benefit the manuscript, and should be addressed by the authors before publication.

Beyond this specific issue, the rest of the reviewer comments are relatively minor suggestions for improvement and ought to be easy for the authors to deal with. However, because the assignment of species to habitat affinity is a primary aspect of the data presented in this paper, and altering it may in fact alter the results, I am returning a decision of major revision. Because of the potential for revised habitat affinity to alter the results or conclusions of the study (even if it does not), I feel compelled to send the revision back to these referees for further review to determine whether your justification for whatever habitat classification you decide to use is sufficient and whether or not the results change if your habitat classification is altered.

·

Basic reporting

The manuscript has clear, unambiguous, and professional English throughout. The literature cited is appropriate and thorough. The article structure and figures and table are professional in quality. The manuscript is self-contained with results relevant to the hypotheses.

Experimental design

The research is within the Aims and Scope of the journal. The research question is well defined and meaningful. The manuscript represents rigorous investigation performed to high technical and ethical standards. The methods are largely described with sufficient detail and information to replicate, except that the details of the data included in phylogenetic reconstruction should be given more detail in the main text, so that the reader does not have to find and evaluate Suppl. Table 1 to evaluate this portion of the methods (which loci, number of nucleotides, etc). Also, on this note, phylogenetic uncertainty should be incorporated in a more rigorous way. For example, statistical comparisons (such as S-H tests) should be conducted to compare resultant tree topologies to each other and to those of prior recent assessments (e.g., Smith and Busby 2014; Knope 2013), as this will obviously effect all downstream inference and the reader should be convinced the phylogenetic hypothesis is rigorous. In contrast to this and a major issue is that the Bayesian MCC phylogeny presented in Figure 2 is not at all similar to Figures 3 and 4, but there is no explanation in the Methods or the figure captions as to why this was done. Of course, changing the relationships amongst species in this way will change the inference of the evolution of body size and reproductive anatomy and parental care, respectively. On this note, it would be best to directly include the phylogenetic uncertainty in the analyses - not just uncertainty in the 95% CI of the ancestral states - but a sensitivity analysis to the inference of character evolution to the alternative tree topologies included in this and the recent studies of the Oligocottinae mentioned above.

Validity of the findings

This study could represent a significant contribution and valuable data to add to our understanding of the evolution of the Oligocottinae, however, as the study is currently designed, it suffers from a critical flaw in how species are assigned to intertidal habitat affinity. The authors make the argument that because all species in the Oligocottinae have been collected previously in the intertidal that they represent an exclusively intertidal radiation with one subsequent reversion in Leiocottus hirundo to exclusively sub tidal habitat affinity. This logic needs to be re-assessed for the following reasons: 1) While species of the genus Artedius can indeed be found in the lowest reaches of the intertidal on very low tides, they are almost exclusively not found in the intertidal and far more common below the reach of tidal fluctuations; 2) to assign species to a habitat affinity based only on their minimum extreme reported depth makes no sense - it is the same as saying the minimum (or maximum) elevation a terrestrial species has ever been found defines its general habitat affinity - some measure of the central tendency (median would likely be a decent choice) of depth would be much more likely to represent the location of the typical habitat of a species; 3) clearly the species of Clinocottus and Oligocottus that are exclusively found in tide pools (compare the average maximum depths for these species to those of Artedius in Table 2 ) do not share the same habitat affinity as members of Artedius that rarely co-occur with these species in most intertidal habitats (simply plotting all depths of collections for each species would demonstrate limited overlap amongst these species groups); and 4) the authors own Figure 2 clearly refutes their assertion that "the Oligocottinae is a primitively and overwhelmingly intertidal group" - in that is shows the maximum depth of the presumed ancestor of the radiation was ~35meters (with 95% CI ~20-50m) - if the group is primitively intertidal, why then is it found in this analysis to have a presumed ancestor with this depth range? Of course, maximum depth is not a good surrogate for typical habitat affinity just as minimum depth is not and therefore all analyses on the relationship of maximum depth to morphological and reproductive characters should be revised to median depth (or some other measure of central tendency) with associated uncertainty in the measure.

Additional comments

Overall, this study represents valuable new data and the writing is clean and clear. The main issue that needs to be addressed before publication is with assignment of species to habitat. I can understand that the authors do not wish to assign species to depth categories (intertidal, transitional, sub tidal as in Ramon and Knope 2008 or Knope and Scales 2013) as depth is a continuous character, but the alternative presented here does not provide a satisfactory alternative. If the authors can amend their analyses to move away from rare collection depths (max and minimum depths reported) and towards better representation of where most individuals of a species are found, their downstream analyses on the relationship between habitat and morphological and reproductive characters in a phylogenetic context will be strengthened and their inferences of character evolution will be more convincing to readers.

Reviewer 2 ·

Basic reporting

Overall, this is a well-written manuscript with a clear logical flow that conforms to the recommended standard sections.

Figures are relevant and high quality. Supplemental files would benefit from having legends to describe them and to define the labels contained within to make them fully interpretably by others. Referencing them in the text will alert readers that they are available and help them identify the step for which they were used.

For example: in Buser_datamatrix.csv include a description of Squamation 1 and 2; and 0=absence/1=presence?

Line 128: Include reference to supplementary file: sculpin_birthdeath_mcc_starbeast.tree
Does the tree file make the full tree, then prune to Oligocottinae + C. pugentensis?

Line 264: Add a caption to Sup Fig 1 to clarify that this is your phylogeny, not the one of Buser & Lopez 2015


Raw data is supplied as supplemental files, identified by a reference to its original source and/or accession number for a relevant public repository. Including the following would aid the reader and clarify the data being analyzed:

Line 103: provide a brief summary of this information here for the convenience of the reader, and as it is contained in the supplementary table of another publication that might not be accessible by all readers.
-names of the 8 loci used
-range of sample sizes/species
-number of outgroups and names or at least an indication of how far away they are (same family/superfamily?)

Line 111: include range of sample sizes (n to n) to define multiple representatives

Line 143: Include a line to specify that the character states for each species are summarized in Table 2.
Orthonopias triacis is missing from Table 2.
Line 272: Specify the cutoff for maximum depth that was used to be classed as “exclusively intertidal” and/or indicate these species in Table 2.

Line 150: How did you choose the max and min depths used in the analysis?
It appears that Sup. Table 2 is additional columns for Sup. Table 1. If so, combine them into a single table (Hint: format one big table in a comma- or tab-delimited text file; easier for others to work with. Or save it as a PDF that readers can zoom in to read; easier to read visually).

How are the consensus depths derived?
Are they consensus across the information in both Sup Table 1 and 2?
Why are some max. consensus depths less than reported maximum depths for a given species?
Where did 8m come from for the min depth for Chitonotus pugetensis, and why is its consensus max half the max depth reported?

What are the references in Sup. Table 3 for? Is there more depth information available than was included in Sup. Tables 1 & 2? Why doesn’t this include the references listed in Sup Tables 1 & 2?

Sup Tables 4-7 appear to be parts of one table; combine into one as suggested above for Sup Tables 1 & 2.

Supplemental Table 8 is a reference list. If these correspond to references in Sup Tables, add them to the end of the appropriate file.


Line 151-2: Did you select the largest of the length measurements reported by different sources to use in the analysis? Do you have a table of these like you do to show the variation in reported depths?

Table 1: Do some museum IDs include more than one individual? For n=8, I expected to see 8 museum IDs.

Experimental design

The quantitative approach described in this manuscript offers an improvement to previous qualitative assessments and provides novel insights on the correlations or lack thereof between habitat, morphology and reproductive traits of oligocottine fishes

As my experience with many of the methods applied is limited or lacking, I defer to the opinions of other reviewers for assessing the subtleties of their implementation.

Validity of the findings

Data accumulated from the literature are referenced and the datasets used here have been made available (but see comments in Basic Reporting).

The authors present conclusions directly relevant to their objective, acknowledge the limitations of their conclusions (e.g., when data is missing or analysis offers weak support) and point out where additional investigation would be beneficial.

Additional comments

Additional minor comments:

Lead author has middle initial ‘J’ on one title page (line 4), but it is missing from the title page that includes the abstract.

The abstract is a well-written summary of the objective, novelty of study and results, but the last statement ( “…and this may be associated with habitat partitioning, particularly as it relates to the degree of wave exposure.”) is not explicitly discussed in the main text.

Line 89, while I appreciate the poetic nature of “muddies the waters”, to improve understanding for the widest audience, replace with “…only further complicates the qualitative categorization of these fishes.”

Lines 117, 168, 201, 467 & 472: Use parentheses around year only

Lines 131-140: Move this information to the introduction.

Line 140: add apostrophe to make possessive: species’ depth preference

Line 221-2 missing end parenthesis and period.

Line 224: reword the subheading to sound less conclusive, similar to the previous subheading, e.g., Depth correlation analysis

Line 262: define MCC

Line 280 & 286: is the “uncertainty of the ML estimates at each node” presented?

Line 290: Sup Fig 2 shows this better than Table 2

Figures 3&4: define the colors/dotted lines along the branches

Line 310: reword slightly: “…0.90), this is the state…”

Line 325: replace These with This: “This finding…”

Line 411: unpaired parenthesis

Line 435: How do these novel body shapes compare to the ‘intertidal stereotypes” in lines 50-52?

Line 437-8 refer to Fig 6 & Sup Fig 3; include a few words here describing what is distinct about the appearance of this subgenus for readers who can’t access Bolin 1944. (e.g., blunt heads and short, stocky bodies referred to in line 444).

Line 480. Include captions for Supplementary figures (e.g. what are each of the lettered insets in SupFig4 and 5?). Best to refer to these earlier in the results or methods to illustrate character states. Avoid presenting new results in the discussion.

Line 481: h missing from “where”

Line 487-8: is an image/illustration of this available, similar to SupFig5

Line 533: slight modification “…correlate with maximum reported depth, the…”

Line 537: slight modification “…between maximum depth and reproductive characters…”

---

## Round 0.2 · Minor Revisions

I am sorry for the delay in getting this back to you, but one of the referees requested additional time to complete their review. We have just now gotten the final review back, and both referees are positive about the value and contribution this work makes to the field. While both feel the work is worthy of publication given the revisions you have made to the manuscript, there are still a number of comments offered by the more critical referee on your revision. The second referee is satisfied with the revisions as they stand, but referee #1 still takes particular issue with your depth distribution data and offers additional feedback to clarify why they feel that the sampling effort and depth at which most individuals are likely to be found is an issue with your analyses.

Ultimately, it may be that you and the referee cannot agree on this issue, but I wanted you to have the opportunity to respond. I am therefore returning the manuscript with a decision of minor revisions to allow you the opportunity to evaluate this additional feedback and make final revisions to the manuscript if you choose. I expect that you will be able to evaluate the feedback and make any final revisions relatively easily, and I look forward to seeing your final manuscript.

·

Basic reporting

Please see comments in "general comments for the author" section.

Experimental design

Please see comments in "general comments for the author" section.

Validity of the findings

Please see comments in "general comments for the author" section.

Additional comments

Reviewer 1 (Matthew Knope)

I appreciate all of the hard work and thoughtful responses the author's have put into my previous comments and believe this manuscript will make an excellent contribution to the field. The revised manuscript is much improved, however, I would like to see a few more modifications (as detailed below) before acceptance.

Experimental design
The research is within the Aims and Scope of the journal. The research question is well defined and meaningful. The manuscript represents rigorous investigation performed to high technical and ethical standards. The methods are largely described with sufficient detail and information to replicate, except that the details of the data included in phylogenetic reconstruction should be given more detail in the main text, so that the reader does not have to find and evaluate Suppl. Table 1 to evaluate this portion of the methods (which loci, number of nucleotides, etc).
We agree with the reviewer’s comments and have added the following text:
123: (sample size per species: 1-9 individuals, median 5)
124-134 from the cottoid families (sensu Smith & Busby, 2014): Agonidae (n = 6 spp.), Cottidae (n = 1 sp.), Hexagrammidae (n = 1 spp.), Psychrolutidae (n = 11 spp.), and Rhamphocottidae (n = 1 sp.). These outgroup taxa are consistent with the most recent phylogenetic hypotheses of broader cottoid relationships (Knope, 2013; Smith & Busby, 2014). This dataset is accessible on Genbank (accession numbers KP826911–KP827632, see Supplementary Table 1) and contains sequence data from eight molecular loci: one mitochondrial protein-coding locus (Cytochrome c oxidase, COI), two nuclear introns [exon-primed intron crossing (EPIC) locus 1777E10 and EPIC locus 4174E20] and five protein-coding nuclear loci [early growth response protein 1 (EGR1); mixed-lineage leukemia (MLL); patched domain-containing protein 1 (ptchd1); Rhodopsin; and Sushi, von Willebrand factor type A, and pentraxin domain-containing 1 (SVEP).
320-321: The trimmed, concatenated MSA dataset spans 4695 aligned nucleotide sites, containing 1037 variable sites.

These are excellent improvements to the Methods.


Also, on this note, phylogenetic uncertainty should be incorporated in a more rigorous way. For example, statistical comparisons (such as S-H tests) should be conducted to compare resultant tree topologies to each other and to those of prior recent assessments (e.g., Smith and Busby 2014; Knope 2013), as this will obviously effect all downstream inference and the reader should be convinced the phylogenetic hypothesis is rigorous.
While we understand the concern of the potential effects of phylogenetic uncertainty on our results, we do not feel that it is necessary to perform a statistical comparison between the phylogenetic hypothesis used in this work with previously published hypotheses (e.g., Smith and Busby 2014, Knope 2013). Comparisons between the phylogenetic hypothesis presented in Buser and Lopez (2015) and previous phylogenetic hypotheses of oligocottine sculpins are discussed at length in Buser and Lopez (2015). The phylogeny presented herein has an identical species topology to that of Buser and Lopez (2015) and an S-H test of two identical topologies is, in our opinion, unnecessary. The dataset assembled in Buser and Lopez (2015) is the most extensive molecular dataset of oligocottine sculpins to date in terms of taxon coverage, sample size per species, geographic coverage of samples, number of molecular loci, diversity of molecular loci, and number of aligned nucleotide sites. These aspects are highlighted and discussed in Buser and Lopez (2015) and we feel that it is redundant and tangential to the current study to repeat this previously published discussion.

Added text:
323-328: As noted in Buser & López (2015), this topology is similar to that of other molecular-based phylogenetic inferences of Oligocottinae (i.e., Ramon & Knope, 2008; Knope, 2013), but has substantially higher support values (i.e., Bayesian posterior probability) for many of the inferred relationships. We will use the classification and taxonomy suggested by Buser & López (2015) for discussion of the interrelationships of oligocottine sculpins.

Fair enough and agreed.

In contrast to this and a major issue is that the Bayesian MCC phylogeny presented in Figure 2 is not at all similar to Figures 3 and 4, but there is no explanation in the Methods or the figure captions as to why this was done. Of course, changing the relationships amongst species in this way will change the inference of the evolution of body size and reproductive anatomy and parental care, respectively.
Figure 2 is a phenogram, so while the representation of the tree is different than in Figures 3 and 4, the relationships between the taxa are unchanged. Rather, the tips and nodes of the phylogeny are positioned in on the vertical axis to reflect the maximum depth or inferred maximum depth (respectively) for each taxon. We have altered the figure caption to more clearly represent what is being conveyed in the figure. Please note that the figures and their order have been altered in the revised version of the text, and the following altered text uses the updated figure listing:

“Figure 4: Inferred evolutionary history of maximum depth and maximum size of oligocottine species. A phenogram showing the inferred evolutionary history of maximum depth is indicated on the left panel. Phylogenetic relationships are represented by white edges and bifurcation points represent inferred speciation events. Phylogenetic topology is from Bayesian MCC tree shown in Figure 3. Relative time is indicated on the horizontal axis and depth in meters indicated on the vertical axis. The tips and nodes of the phylogeny are positioned on the vertical axis to reflect the maximum depth or inferred maximum depth (respectively) of each taxon. Ancestral states for each node were inferred using maximum likelihood and 95% confidence intervals for each state are represented with blue lines. Species names are abbreviated as follows: Artedius corallinus = Ar. cor., A. fenestralis = Ar. fen, A. harringtoni = Ar. har., A. lateralis = Ar. lat., A. notospilotus = Ar. not., Chitonotus pugetensis = Ch. pug., Clinocottus (Oxycottus) acuticeps = Cl. acu., Clinocottus (Clinocottus) analis = Cl. ana., Clinocottus (Blennicottus) embryum = Cl. emb., C. (B.) globiceps = Cl. glo., C. (B.) recalvus = Cl. rec., Leiocottus hirundo = Li. hir., Oligocottus maculosus = Ol. mac., O. rimensis = Ol. rim., O. rubellio = Ol. rub., O. snyderi = Ol. sny., Orthonopias triacis = Or. tri.”

Thank you for this clarification and for the improvement in clarity of the figure legend.

On this note, it would be best to directly include the phylogenetic uncertainty in the analyses - not just uncertainty in the 95% CI of the ancestral states - but a sensitivity analysis to the inference of character evolution to the alternative tree topologies included in this and the recent studies of the Oligocottinae mentioned above.
The two alternate phylogenetic studies of sculpins highlighted by Reviewer 1 (i.e., Knope 2013, Smith and Busby 2014) are broad-scale phylogenies with limited intra-generic (Smith and Busby 2014) or intra-specific (Knope 2013) sampling. Smith and Busby, for example, include only 4 (out of 16) oligocottine species in their analysis. While Knope (2013) is based on only two loci (~1200 aligned nucleotide sites, 800 of which is mtDNA), contains only a single representative of each species, and has only moderate support for phylogenetic relationships among oligocottines. I have embedded the pertinent subtree from Knope (2013) for your reference, and highlighted in red the nodes with less than 0.80 Bayesian posterior probability, which is 9/15 nodes within Oligocottinae (as compared to 3/15 nodes with <0.95 for the phylogeny used herein, see updated Fig. 3).
While these studies have each provided valuable insight into broader sculpin relationships, with respect to the interrelationships of oligocottine sculpins, neither provide a meaningful alternative to the phylogeny used in this study (e.g. Buser and Lopez (2015) dataset contains all 16 oligocottine species and many outgroups, 8 molecular loci, 4695 aligned nucleotide sites, 650 of which are mtDNA, median 5 individuals per species, broad geographic sampling, generally high node support values) and we do not see a benefit or potential for additional insight into our study objectives by using the alternative phylogenetic hypotheses suggested by Reviewer 1. There are other phylogenetic hypotheses of Oligocottinae based on, e.g., expertise/intuition (Bolin, 1947), larval characters (Washington, 1986), or morphology (Strauss, 1993), but here too, the phylogeny used in this study is much more robust and we do not see the potential for additional insight using these alternative topologies.
As for phylogenetic uncertainty represented by the phylogenetic hypothesis used here, the signals in our data are generally very strong (e.g., all oligocottines are commonly found in tide pools with the exception of 1 species, Leiocottus hirundo), and the support values in our phylogeny are generally high (as noted above). Our conclusions are based only on those nodes with high support and very clear signal in the data.

Agreed.

Validity of the findings
This study could represent a significant contribution and valuable data to add to our understanding of the evolution of the Oligocottinae, however, as the study is currently designed, it suffers from a critical flaw in how species are assigned to intertidal habitat affinity. The authors make the argument that because all species in the Oligocottinae have been collected previously in the intertidal that they represent an exclusively intertidal radiation with one subsequent reversion in Leiocottus hirundo to exclusively sub tidal habitat affinity. This logic needs to be re-assessed for the following reasons:

1) While species of the genus Artedius can indeed be found in the lowest reaches of the intertidal on very low tides, they are almost exclusively not found in the intertidal and far more common below the reach of tidal fluctuations;
Despite extensive study of museum records (detailed below), and substantial personal observation of many of these taxa in the field (noted in Buser and Lopez 2015), we are unable to verify this assertion about the depth distribution of Artedius species by Reviewer 1. Personal experience, museum data, primary literature, and field guides all indicate that Artedius species are found with some regularity in tide pools and intertidal areas. In fact, museum records show that A. harringtoni and A. lateralis are recorded from intertidal areas (i.e., depth = 0 m or explicit mention of tide pool or intertidal collection locality) with more frequency than they are from all other depths combined (see update Figure 1, Supplementary Tables 2 and 4). The recorded depth of the remaining Artedius species shows that intertidal areas make up 26% (A. corallinus and A. notospilotus) to 45% (A. fenestralis) of collection localities (Supplementary Table 4). While the phrase “almost exclusively not found in the intertidal” does not carry with it an explicit criterion, we strongly assert that the documented distribution of species of Artedius do not meet even the most liberal interpretation of this description.

I think at issue here is the reliance on data sources for depth that are not based on similar effort, detectability, or methodology and these issues compromise all downstream analyses based on depth. For example, intertidal ichthyofauna surveys are likely much more commonly conducted than nearshore subtidal ichthyofauna surveys and, in addition, intertidal, shallow subtidal, and deep-water surveys are all conducted with radically different methodology and effort (e.g., frequent surveys with dipnets at low tide vs. less frequent SCUBA collections vs. ship-board trawl surveys). While some reassurance is apparently gained from congruence between depth ranges in previously published works (Bolin, 1944; Miller & Lea, 1972; Eschmeyer, Herald & Hammann, 1983; Mecklenburg, Mecklenburg & Thorsteinson, 2002) and museum lots being not statistically significantly different from one another (your phylogenetic paired t-test), these distributions are not statistically independent in that many of the same collections that comprise the museum lots are what were used to assign depth distributions in these publications (as just one example, Bolin’s 1944 collections are housed at the California Academy of Sciences) and this should be addressed.

2) to assign species to a habitat affinity based only on their minimum extreme reported depth makes no sense - it is the same as saying the minimum (or maximum) elevation a terrestrial species has ever been found defines its general habitat affinity - some measure of the central tendency (median would likely be a decent choice) of depth would be much more likely to represent the location of the typical habitat of a species;
If the depth distribution of each species was bell-shaped, we would agree with Reviewer 1’s assertion that taking the extreme ends of the distribution would be inappropriate. However, plots of the distribution of collection depths for museum specimens of each oligocottine species show that these distributions are not bell-shaped (updated Figure 1). It also shows that for each oligocottine species except L. hirundo, at least 26% of collection depths are intertidal (i.e. 0 m depth) (range: 26-95 %, see Supplementary Tables 2 and 4). In fact, the median collection depth is 0 meters for 11/16 oligocottine species (Supplementary Table 4). So, while the minimum collection depth does represent an “extreme” end of the distribution in the literal sense, it is a common (often VERY common) collection depth for all oligocottine species except L. hirundo. We do not currently know with certainty the general habitat affinity of these species, but given the high frequency with which they are collected from tide pools and intertidal areas, it is at least defensible to conclude that intertidal residence is a realized life history strategy for all but one member of the group. What is interesting, however, is that there appears to be variation in depth residency across members of the group, as evidenced by the distribution of collection depths for each species. While some species appear to have been collected exclusively from very shallow depths (0-1 m, e.g., Clinocottus (Blennicottus) embryum), others have been collected from a wide range of depths (e.g., Artedius notospilotus, see Figure 1). A wide range of collection depths suggests that there is a variety of suitable depths for those species. We thus conclude that a single metric (e.g., median depth) is an overly-simplified representation of the ecology of these species. We assert that a minimum and maximum depth for each species is far more informative. To avoid the influence of outliers on the characterization of the depth range of each species, we selected a depth range that includes 95% of museum collection depths for each species. For species with a high (>25%) occurrence of intertidal collection depths, we set 0 m as the minimum and selected the shallowest depth that includes 95% of museum collections as the maximum (see Figure 1, Table 1, Supplementary Tables 2 and 4). For L. hirundo and the outgroup taxon Chitonotus pugetensis, we discarded the shallowest 2.5% and the deepest 2.5% of collection depths and used the resultant minimum and maximum collection depths (see Figure 1, Table 1, Supplementary Tables 2 and 4).
We re-ran our analyses using the 95% museum collection depth range and also re-ran our analyses using median depth, and found, with one exception, no change in the outcome of any of our tests with either alternative depth classification. E.g., body shape vs depth:

As you can see, while a few details change between the depth characterizations, the overall picture is the same, even using median depth. The differences between the depth range reported in previously published works (i.e., the depth range we originally used) and the 95% depth range of museum lots are not statistically significant (phylogenetic paired t-test p-value > 0.89) and, predictably, the outcomes of using either of the two depth ranges are virtually identical. The exception to this is only minor: we reported in our original text that length and maximum depth have a relationship that is significant, but just barely so. With the updated depth classification, the relationship between length and depth is no longer significant. We were cautious in our original interpretation of the depth covariate, so the changes to the text are only minor. We’ve updated the text to convey these ideas, and given the more verifiable nature of the museum records, adopted the 95% depth range of museum lots as our means of depth range characterization. We also added an additional character “Tide pool occupancy (Presence, absence)” to characterize those taxa that are commonly found in tide pools.

The author’s assert, “If the depth distribution of each species was bell-shaped, we would agree with Reviewer 1’s assertion that taking the extreme ends of the distribution would be inappropriate.” This response demonstrates confusion regarding my original comment. Regardless of the underlying actual depth distribution for each species, using only the extremes of the range is likely to fail to capture the depth that most individuals within each species occupy, unless the abundance data is extremely left- or right-skewed. The analyses above perpetuate the use of data to describe the distribution based on methods that likely have extreme sampling bias and non-independence (as explained in comments above) and this should be made explicitly clear to the reader.

Added and modified text:
169- 197: Collection data for all specimens of each species of Oligocottinae and the outgroup taxon C. pugetensis were collated from museum records from the following natural history collections: University of Alaska Museum (UAM), University of British Columbia Beaty Biodiversity Museum (UBCBBM), University of Washington Burke Museum Fish Collection (UW), Oregon State University Fish Collection (OS), California Academy of Sciences (CAS), Natural History Museum of Los Angeles County (LACM), University of Michigan Museum of Zoology (UMMZ) and Scripps Institute of Oceanography Marine Vertebrates Collections (SIO). These records were accessed through institution-specific (UW, UBCBBM, CAS) or the multi-institutional database interfaces (all others) VertNet.org, Arctos.Database.Museum, and FishNet2.org (see Supplementary Table 2 for all museum records analyzed). For each species, we extracted collection depth data from all museum holdings of adult specimens for which it had been recorded. Some collection depths are recorded as a range, in these cases, we used the maximum depth in the range. Where the collection depth and/or locality is described as “tide pool,” “intertidal,” etc., we assigned a collection depth of 0 m. To lessen the effects of outliers, we selected a depth range (i.e., minimum depth and maximum depth) for each species that includes 95% of museum collection depths (illustrated in Figure 1). For the purposes of this study, we will refer to this depth range as the range where each species is “commonly” collected. To verify these depth ranges, maximum and minimum depth records for each species were cataloged and cross-examined from multiple sources (Bolin, 1944; Miller & Lea, 1972; Eschmeyer, Herald & Hammann, 1983; Mecklenburg, Mecklenburg & Thorsteinson, 2002; see Supplementary Table 3). Where these previously published depth maxima and minima disagree, we chose the median value for each. Many of these ranges include only imprecise descriptions such as “tide pools” and “intertidal areas.” In these cases, we assigned a minimum depth value of 0 m and a maximum depth value of 2 m. We used a phylogenetic paired t-test (Lindenfors, Revell & Nunn, 2010) to compare the maximum and minimum depth for each species using the museum records vs. the descriptions published in the literature using the “phyl.pairedttest” function in the R package “phytools” (Revell, 2012; see “Character coding- Depth Range” section in “LitorallyAdaptiveScript.R” in Supplementary Folder “LitorallyAdaptive_PeerJ_Rfolder”).

198-200: Tide pool occupancy (Presence, absence). We noted which taxa were explicitly collected from tide pools in museum collection data, in previously published depth ranges, and in primary literature.

280-284: For both museum records and previously published depth ranges, preliminary results indicated that, while there is considerable variability in the maximum collection depth of each species in Oligocottinae, all species share a minimum recorded depth of zero meters. Given this invariability in minimum depth, we chose to use only maximum depth as our depth variable for regression analysis.

329-353: The outgroup taxon, C. pugetensis, rarely (if ever) occurs in intertidal areas (Fig. 1, Table 1, Supplementary Tables 3 & 4). However, apart from L. hirundo, all the constituent species of Oligocottinae are regularly found in intertidal habitats and both museum records and published depth ranges include tide pools in the common collection depth or depth range data for all oligocottine species but L. hirundo (Fig. 1, Table 1, Supplementary Tables 3 & 4). There is also explicit discussion of tide pool and intertidal occupancy for all oligocottine species except L. hirundo in the primary literature (Supplementary Table 3). However, while the occupation of intertidal and subtidal habitats is often portrayed as an either/or scenario, there is considerable variation in the maximum depth at which each species occurs (Fig. 1, Table 1). Generally though, all oligocottine species occur at relatively shallow depths: none is commonly collected at depths greater than 55 m, most (12/16 spp.) are not commonly collected below 25m (though there is some discrepancy between the museum collection data and the published depth ranges for A. corallinus and A. fenestralis), and four (published ranges) to seven (museum depth data) species are common only in very shallow (i.e., 2 m depth or less) habitats (Table 1). There is considerable disagreement between the museum collection data and the published depth range for A. notospilotus, C. acuticeps, C. analis, and L. hirundo. In each case, published depth ranges indicate a maximum depth that is > 10 m deeper than the depths where these species have been commonly collected in museum holdings (Supplementary Tables 3 and 4). However, the depth ranges are otherwise largely congruent, and the differences between the two datasets are not statistically significant (phylogenetic paired t-test p-value > 0.89). All remaining analyses show identical outcomes when using either the common museum collection data or the previously published depth range data for each species. Given the congruence of the datasets, the indistinguishable outcome of using one over the other, and the more verifiable nature of the museum collection records, we present the results of the remaining analyses using only the common museum collection depth range of each species.


355-368: (MRCA) of Oligocottinae likely occurred in shallow habitats (ML estimate: 1 m; 95% confidence interval: 0 m, 2 m). Ancestral state reconstruction of tide pool occupancy shows that with extremely high proportional likelihood (0.9988) the MRCA of Oligocottinae occurred in tide pools. In fact, even the MRCA of the Leiocottus lineage was likely (0.9215 proportional likelihood) capable of living in tide pools (Supplementary Figure 1). Thus, the absence of tide pool occupation in L. hirundo likely represents a derived state. The ASR of maximum depth suggests that the MRCA of Oligocottinae occurred down to only moderate depths (ML estimate: 23 m; 95% confidence interval: 2 m, 44 m; see Fig. 4) and suggests that the habitation of only very shallow-water habitats (maximum depth = 2 m or less) seen in members of Oligocottus maculosus, O. rimensis, and O. snyderi and in all members of the subgenus Clinocottus (Blennicottus) represents a derived state (see Table 1, Fig. 4). However, given the uncertainty of the ML estimates of maximum depth at each node (Fig. 4), and the uncertain phylogenetic relationships of Blennicottus, Leiocottus, and Oligocottus lineages (Fig. 1), it is not possible to claim with confidence the number of transitions that may have occurred within the subfamily.

454-455: No morphological, reproductive, and body shape variables examined in this study show a significant correlation with maximum depth.

489-492: Small maximum size and a reduction in scales have been noted as common features of intertidal fishes by previous authors (Gibson, 1982; Knope & Scales, 2013), and while we found no evidence to support these hypotheses within Oligocottinae, oligocottines as a whole may in fact offer support.

Knope and Scales (2013) Table 1 reports number of scales for each of these taxa and those data should be used rather than presence/absence of scales in a regression of depth to scales. On this note, I believe generalized least squares analysis between depth (a continuous character) and presence/absence of scales (a categorical character) is inappropriate. With a continuous predictor and a categorical response variable you want to use a logistic regression, but this is a moot point, if using the data from Knope and Scales.

3) clearly the species of Clinocottus and Oligocottus that are exclusively found in tide pools (compare the average maximum depths for these species to those of Artedius in Table 2 ) do not share the same habitat affinity as members of Artedius that rarely co-occur with these species in most intertidal habitats (simply plotting all depths of collections for each species would demonstrate limited overlap amongst these species groups); and
This characterization of the depth distributions of oligocottine species is not supported by the plots of collection depths suggested by Reviewer 1 (see updated Fig. 1). There is substantial overlap between collection depths of all oligocottine species (excepting perhaps L. hirundo), especially between Artedius lateralis, A. harringtoni, A. fenestralis, all Oligocottus species, and Clinocottus analis. The collections data analyzed in this study is representative of the full geographic distribution of each species and comes from eight natural history museums (University of Alaska Museum, University of British Columbia Beaty Biodiversity Museum, University of Washington Burke Museum, Oregon State University, California Academy of Science, Natural History Museum of Los Angeles County, Scripps Institute of Oceanography, and the University of Michigan Museum of Zoology). We know of no other sources of data that would invalidate or dispute our conclusion on this matter.

Again, please see comments above regarding the use of these museum data to characterize abundance across depth range. In addition, to assign all collections described as “intertidal” or “tidepool” to 0 meters is almost certainly why it appears that all species of Artedius have full overlap with all species of Clinocottus and Oligocottus at most collection sites. Tides throughout the range of these species often vary by 3 meters or more, and ignoring this variance in the assignment of collections in the intertidal all to 0 meters, will automatically collapse all species occurrences in the intertidal together and this should at least be acknowledged in the text.

4) the authors own Figure 2 clearly refutes their assertion that "the Oligocottinae is a primitively and overwhelmingly intertidal group" - in that is shows the maximum depth of the presumed ancestor of the radiation was ~35meters (with 95% CI ~20-50m) - if the group is primitively intertidal, why then is it found in this analysis to have a presumed ancestor with this depth range? Of course, maximum depth is not a good surrogate for typical habitat affinity just as minimum depth is not and therefore all analyses on the relationship of maximum depth to morphological and reproductive characters should be revised to median depth (or some other measure of central tendency) with associated uncertainty in the measure.
Reviewer 1’s confusion at our concluding remarks appears to stem from a misunderstanding about how we are characterizing the habitat of each species. While “primary” habitat is often portrayed as a singular, fixed state (e.g., “intertidal”), the data we present show that for many species, the habitat is better characterized as a range of potential depths. Among those potential depths however is one particular habitat (i.e., intertidal), which requires a prerequisite suite of adaptions for it to be viable. Given the frequency with which these species have been collected in the tide pools (updated Supplementary Tables 2-4), we conclude that intertidal habitats present a viable habitat state for all oligocottine sculpins except L. hirundo. Predictably, the ancestral state reconstruction (ASR) shows that the presence of tide pool occupancy is overwhelming likely (proportional likelihood 0.9988) for the MRCA of Oligocottinae, and that minimum depth of the MRCA of Oligocottinae has a maximum likelihood value of 1 m (95% confidence interval: 0, 2 m). So, while the ancestral state reconstruction shows that the maximum depth of the MRCA of Oligocottinae has a maximum likelihood value of 23 m, that does not negate the minimum depth estimate, it means only that the MRCA likely had a broad range of potential habitable depths (much like many extant oligocottines), which likely included the intertidal and tide pools. If that is the case, then the narrow range of depths seen in some oligocottine species likely represent a derived state, but the ability to live in intertidal habitats is a primitive state, and it is present in all but one species (i.e., the group is overwhelmingly capable of living in the intertidal). We discussed this explicitly in the original text in lines 393-402 but we can see how the phrase, “Oligocottinae is a primitively and overwhelmingly intertidal group,” could lead to confusion. We have updated the text to be as clear as possible:

Added and modified text:
Lines 28-29: the ability to live in intertidal habitats, particularly in tide pools, is likely a primitive state for Oligocottinae

478-488: The subfamily Oligocottinae should be thought of as a clade of intertidal-occurring fishes and the ability to live in intertidal depths and specialized intertidal habitats such as tide pools is likely the ancestral state of the group. This finding does not support the hypothesis that there is differential diversification of intertidal vs. subtidal oligocottine groups (e.g., Ramon & Knope, 2008; Knope & Scales, 2013), as we conclude that virtually all oligocottines reside with some frequency in intertidal habitats. However, this ability to live in tide pools does not preclude residency in other habitat types within the same species, as many of the extant and ancestral species are capable of living in a variety of depths in addition to the intertidal ones. Thus, the diversification of Oligocottinae should not be characterized as occurring between intertidal and subtidal habitats, but rather occurring within a habitat range that includes both. This may explain the general lack of correlation between depth the other characters examined in this study.

I appreciate the author’s clarification on these issues. However, I would like to direct the authors to the text of Ramon and Knope (2008) and Knope and Scales (2013) to clarify some further apparent confusion. Both of these papers assign some members of the Oligocottinae to occupation of just the intertidal and some to both the intertidal and the subtidal (See Fig. 1 in Ramon and Knope and Fig. 2c in Knope and Scales), as is the case in this manuscript. The selected passages below show that both of these papers place the transition from exclusively subtidal habitat affinity to occupation of the intertidal, including a broad range of depths into the subtidal for some species, as basal to the Oligocottinae and this should be corrected in the manuscript and not presented as a novel finding. For example, lines 515-518, “We thus conclude the opposite of previous studies and suggest that rather than containing an intertidal radiation (Ramon and Knope 2008; Knope and Scales 2013), the subfamily Oligocottinae itself represents an intertidal radiation.”

From Ramon and Knope (2008) Discussion 4.2 (p. 481; please see paper for references cited herein):

“Mapping habitat affinities onto a phylogeny can provide a fuller understanding of the evolutionary patterns within a clade (Ahn and Ashe, 2004; Bargelloni et al., 2000; Ruber et al., 2003; Williams and Reid, 2004). When we mapped habitat affinities onto our molecular phylogeny, a very clear trend appeared where ancestral species are found primarily in the subtidal and derived species are primarily intertidal (Fig. 1). All species basal to the A–C–O clade are subtidal species, with the exceptions of L. armatus (intertidal and freshwater), H. hemilepidotus (subtidal and intertidal) and E. bison (subtidal and intertidal). While we did not sample all species in the branch considered to be immediately basal to the Oligocottinae, the entire clade is a deep-water subtidal branch (Bolin, 1947).
Within the A–C–O, the Artedius species can be found both in the intertidal and subtidal. However all of the Oligocottus and Clinocottus species are intertidal with the exceptions of two species (Oligocottus rubellio and C. acuticeps) found in the intertidal as well as in other habitats (subtidal and freshwater). Martin (1991) and others (Wright and Raymond, 1978; Yoshiyama and Cech, 1994) have found that several resident intertidal species (Oligocottus maculosus, Oligocottus snyderi, C. analis, C. globiceps, and C. recalvus) have specific physiological (air-breathing) and behavioral traits not present in the ancestral subtidal species (J. zonope, Icelinus borealis and Chitonotus pugetensis), which further supports our hypothesis of a subtidal to intertidal invasion. However, Martin (1996) is the only study that has tested for air-breathing ability using depth as an ecological gradient in fish. Clearly, more studies like this are needed to conclude that air-breathing ability is an adaptation to the intertidal and not the result of an ancestral trait in subtidal species. We are, of course, not the first to suggest that subtidal species invaded the intertidal, however our phylogeny is the first that we are aware of to clearly support the hypothesis that subtidal sculpin species are ancestral to intertidal sculpin species.”

From Knope and Scales (2013) Discussion (p. 480; please see paper for references cited herein):

“First, basal-rooted species of marine sculpins along the North American Pacific Coast are found predominantly in the subtidal and the shift to the intertidal resulted in diversification (Ramon & Knope, 2008; Mandic et al., 2009a; Knope, 2013), giving rise to at least the seven transitional and nine intertidal species examined in this study (Fig. 2c).”

Reviewer 2 ·

Basic reporting

no comment

Experimental design

no comment

Validity of the findings

no comment

Additional comments

I am satisfied with the authors’ responses to my previous comments. Their revisions clarified the areas I previously found confusing, resulting in a stronger manuscript supported by well described supplementary materials.

---

## Round 0.3 · accepted · Accept

I have read your responses and can appreciate both sides of this disagreement. Despite the fact that you and one of the referees do not agree on every point, I feel that you have made a defensible case for how and why you have conducted your study this way, and the other referees have been satisfied with your approach. As you point out, the inclusion of the review history will allow anyone interested to see the debate and the responses, so I am happy to move your article forward into production at this time.